



# Analysis of the Campinas tornado (Brazil) in June 2016: damage track, radar characteristics and lightning observations of the supercell

Lucí Hidalgo Nunes1*, Gerhard Held 2*, Ana Maria Gomes 2*, Kleber Pinheiro Naccarato3, Raul Reis Amorim[1]

[1]Department of Geography, Institute of Geosciences, State University of Campinas, Unicamp, Campinas, Brazil

[2]Meteorological Research Institute, São Paulo State University, Unesp, Bauru, Brazil

[3]Division of Impacts, Adaptation and Vulnerabilities, National Institute for Space Research, INPE, São José dos Campos, Brazil

[*] Retired

*Correspondence:* lhidalgo@unicamp.br

**Abstract.** Shortly after midnight on 05 June 2016, in the city of Campinas with >1.2 million inhabitants, located in the State of São Paulo, intense precipitation, including hail, a large number of electric discharges and very strong wind gusts, causing significant damage, were recorded. No fatalities were documented, probably due to the day and time (Sunday around 00:20 Local Time). The affected areas are middle and even upper-middle class neighborhoods, with solid buildings, confirming the potency of the phenomenon. The destruction pattern indicates an intense perturbation resulting in the twisting of structures and tree branches, as well as large objects having been airborne over a distance of about 50 m, and large trees ripped from the ground, all suggesting that it was a tornado of category EF2-3. Severe damage was also reported from other towns in the region. About three hours before the tornado occurred in Campinas, an even stronger event devastated part of the small town of Jarinu, 40 km southeast of Campinas, possibly an EF3 tornado, which caused one fatality and overturned two semi-trailer trucks. No alerts that a disturbance of this magnitude would impact the region were raised, demonstrating that Campinas, and probably most other Brazilian cities and towns, are not prepared for such an event.

During the beginning of June 2016, the synoptic situation over Brazil was characterized by a strong anticyclone centered over the northern half of South America at the 250 hPa level and bounded by a strong zonal Subtropical Jet (STJ) in the south, resulting in moist air being advected from the Amazon and Pacific region, creating favorable conditions for strong convection in the State of São Paulo, even during the night.

Images from a Doppler S-band radar, located in Bauru, recorded a supercell storm lasting 8.5 hours, which traversed the eastern half of the State of São Paulo during the night of 04/05 June 2016 and spawned a tornado in the city of Campinas during the early hours of the morning (Local Time). Despite the distance of >200 km, these radar observations confirmed typical tornado signatures, such as a rotational damage pattern, a hook echo and a mesocyclone with a rotational velocity of 12.5 m s-1. The supercell was accompanied by intense lightning activity throughout its life cycle with a "lightning jump" from 0 to 55 ground strokes per minute within 12 min just prior to the tornado touch-down, culminating in a frequency of 238 strokes per minute of Total Lightning. Although some of the Severe Storm



Parameters calculated by the TITAN (Thunderstorm Identification, Tracking, Analysis and Nowcasting) Software were slightly lower than found in previous tornado cases in the State of São Paulo, this is most likely due to the fact that this was the first occurrence of a tornado observed by radar during the dry austral winter season in this region of Brazil, as well as a nocturnal event.

## 1 Introduction


Tornadoes are the most intense vortices in the atmosphere over land, formed in environments with intense wind shear. They are associated with conditions of great thermodynamic instability, reinforced by specific surface parameters, such as relief, vegetation, urbanization and the presence of water bodies. There are several scales to classify tornadoes by damage, such as F (Fujita, 1981), later officially modified to EF (Enhanced Fujita; National Weather Service, 2020),

Torro, developed in England by H. E. Brooks (TORRO, 2020), and EBRAV, emerging in Brazil (Candido, 2012). All these scales are based on the degree of destruction brought about by tornadoes, but since the characteristics of buildings and building materials are different across countries (for example, in the USA motor homes are commonly used, while in the plains of North America timber buildings prevail, less common in much of Brazil), the comparison among nations is challenging.

Tornadic cells can develop in atmospheric environments that produce severe weather conditions such as transient frontal systems, mesoscale convective complexes, cyclones, local convective systems and supercell storms. They are not associated with a single weather condition and thus, occur in different climates and during different periods of the year. However, the most severe tornadoes are associated with supercells, which have an extremely intense upward rotating current, reaching speeds of 50 m s-1 or more (Houze, 1993).

The rising number of detected tornadoes could be associated with the increase and spread of the world's population, as well as greater awareness of such phenomena (Doswell et al., 1999), because the rapid dissemination of news about severe events is increasingly facilitated. No association between the frequency of tornadoes and global warming has so far been confirmed and requires further studies. Rosenfeld et al. (2008) claim that the higher concentration of aerosols tends to intensify convection in the atmosphere, generating deeper clouds and thus, more suitable conditions to produce

severe events, such as tornadoes. This process is related to reducing the average size of water droplets within the cloud; being lighter, the droplets tend to reach higher parts of the cloud, where they freeze and, in the process, release more latent heat, boosting convection at medium and high levels. Due to the higher concentration of anthropogenic pollutants, large urban and industrial centers would be conducive environments for this type of intensification and, therefore, would contribute to the formation of more severe storms. This was also confirmed indirectly by Naccarato et al. (2003),

who found significantly increased lightning flash densities over large cities in the State of São Paulo.

Relatively few in-depth studies of atmospheric conditions associated with tornadoes in Brazil have been conducted. However, observations of tornadoes and waterspouts (when occurring in oceans or large water bodies such as lakes) have been reported regularly in various media in recent years, but probably also occurred in the past, as attested by the reports collected by Candido (2012). In a survey and analysis of tornadoes in Brazil between March 1877 and April

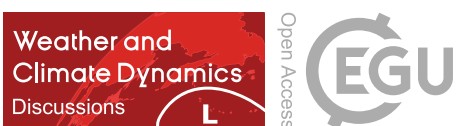

2011, Candido (2012) found 205 episodes. He also noted that just as in other parts of the world, in Brazil tornadoes are more common in the fall and spring, which are seasonal transition periods with more atmospheric instability. Other authors have studied the occurrence of tornadoes in different parts of Brazil with radar information (Bertoni, 2013; Vilar, 2016) while some others were developed without radar information (Marcelino, 2003; Marcelino et al., 2003; Nunes et al., 2008; Insee et al., 2016).

According to historic media coverage, Campinas and the surrounding region experienced devastating tornadoes in 1995 and 2001, but these events had not been scientifically evaluated. At the end of November 1995, a tornado with winds of about 180 km h-1 hit the campus of the State University of Campinas (UNICAMP), destroyed the 250 ton. roof of the sports gym, broke windows of several buildings and uprooted 12 trees on the campus, as well as caused considerable damage in other parts of the city (Folha de São Paulo, 1995). During late afternoon of 4 May 2001, a tornado with winds of up to 300 km h-1 hit nine towns in the Campinas region, caused general destruction, and left several persons injured, as well as at least one fatality (Folha de São Paulo, 2001).

The first tornadic cell fully documented by radar in Brazil occurred in 1994 within the range of the Bauru S-band Doppler radar (Gomes et al., 2000). Several more events could be tracked in the State of São Paulo from 2004 onwards, when already lightning observations were available, with the objective to identify characteristic tornado signatures in the radar images and lightning data which could be used for Nowcasting (Held et al., 2005, 2006a, b, c, 2009, 2010a, b, 2011, 2014; Gomes et al., 2009; Nascimento et al., 2014). Some of the studies also attempted to investigate numerical forecast models to identify parameters which could indicate the possible occurrence of severe storms likely to generate a tornado up to three hours in advance (Held et al., 2005, 2006b, 2009, 2014; Nascimento et al., 2014).

Since 1994, several events of supercell storms traversing the State of São Paulo while spawning tornadoes, although relatively rare, had been observed and tracked by the Doppler S-band radars located in Bauru and Presidente Prudente, respectively, (Held et al., 2010a). The majority of these storms, which were analyzed in detail, had spawned severe tornadoes and occurred during the atmospheric transition periods (May and September). The timing of the transition periods varies slightly from year to year, depending on the large-scale circulation in the troposphere. However, mostly less severe occurrences, with several ones only generating funnels that did not reach the ground, were also observed during the subtropical "Rain Season" (October - March), e.g., near Lins (a small town in the central State of São Paulo) on 25 December 2008 (Gomes et al., 2009). Mean monthly rainfall totals in the central interior of the State of São Paulo remain well below 50 mm from June to August (INMET, 2019; IPMet, 2019; Held and Nachtigall, 2002), thus this period can be classified as the dry austral winter period. Severe storms associated with strong winds are quite uncommon during this period, although severe hailstorms, mostly associated with baroclinic events, had been recorded occasionally in the interior of the state (Gomes and Held, 2009).

This study presents the analysis of the life cycle of a supercell which spawned a tornado at around 00:20 Local Time (LT) on 05 June 2016, causing severe damage in parts of Campinas, a Brazilian city in the State of São Paulo that houses more than 1 million inhabitants. The day (Sunday) and time of the tornado occurrence probably contributed to the fact that only a small number of persons suffered injuries, without any fatalities having been reported. However, based on the damage pattern, it could be classified as EF2-3 on the Enhanced Fujita scale, which since 2007 replaced



the original Fujita Scale (Fujita, 1981). Both scales classify tornadoes from 0 to 5 based on the damage observed after the passage of a tornado, but the EF scale incorporates more damage details (National Weather Service, 2020).

Nearby municipalities, located within a 50 km radius from Campinas (marked in Fig. 6), also recorded heavy rains and strong winds during the night, e.g., Morungaba (28 km east of Campinas; in the extended path of the tornadic supercell),
but others were affected by different storm systems, such as Itupeva, Atibaia and Jarinu. In the latter town an earlier supercell peaked at around 21:50 LT on the previous day and based on damage and eyewitness reports also spawned a tornado estimated to have been an EF3. However, being just outside the Bauru 240 km quantitative radar range, only a qualitative description of observations is included.

## 2 Data and methods

The analysis of this event is primarily based on a field survey of the affected areas, performed during the days immediately following the occurrence of the tornado, in order to determine damage patterns *in loco*, on the interpretation of data from the Doppler S-band radar in Bauru (BRU; 22.36º S, 49.03º W) operated by the *Centro de Meteorologia de Bauru* (CMB) of the *Universidade Estadual Paulista* (UNESP), as well as on lightning stroke data obtained with the Brazilian Lightning Detection Network (*Rede Brasileira de Detecção de Descargas Atmosféricas*
BrasilDAT) operated by the *Divisão de Impactos, Adaptações e Vulnerabilidades* (DIIAV) of the *Coordenação-Geral de Ciências da Terra* (CGCT) of the *Instituto Nacional de Pesquisas Espaciais* (INPE). Additional radar data from the Doppler S-band radar in Presidente Prudente (PPR), also operated by the CMB, as well as from the S-band radar at São Roque (SRO), operated by the *Departamento de Controle do Espaço Aéreo* (DECEA), are supplementing the study of convective activity for this case. It must be pointed out that the SRO is an operational radar, primarily deployed for Air
Traffic Control, and thus not very suited for research, resulting in unreliable data quality (many VOL-scans are incomplete, due to missing higher elevations, as well as individual rays sometimes missing). Furthermore, no radial velocities were available for this case. Also, a difference of 10–15 dBZ between BRU and SRO was noted, and therefore the authors decided to only use these data in a qualitative manner.

Volume scans of the original radar data (BRU and PPR), recorded and archived in the propriety format *Interactive*
*Radar Information System* (IRIS) of SIGMET Inc. (SIGMET, 2005), were converted to the MDV format (Meteorological Data Volume), and subsequently processed with the software "TITAN" (Thunderstorm Identification, Tracking, Analysis and Nowcasting; Dixon and Wiener, 1993), available from NCAR (National Center for Atmospheric Research, Boulder, Colorado, USA). The CMB radars have a 2° beam width and each volume scan with 16 elevations is completed in 7.5 min within the 240 km quantitative radar range. The range extends to 450 km in surveillance mode
(PPI only at 0.3° elevation). The SRO radar also has a 2° beam width, but each volume scan with 15 elevations is initiated every 10 minutes. The conversion of the raw data (HDF5 format) to the MDV format was performed up to 250 km.





TITAN is a Software System which produces a variety of important parameters for a chosen reflectivity and volume
threshold throughout the lifetime of storms, such as Area, Volume, Precipitation Flux, VIL (Vertically Integrated Liquid
water content), Maximum Reflectivity, Hail Metrics, speed and direction of propagation, etc, per volume scan, as well
as cell tracking, including splits and mergers of cells. It also has the facility to collocate lightning flashes with the radar
echoes, including a separation into positive and negative cloud to ground strokes (CG). For this analysis TITAN was
running with a resolution of 750 m (also 250 m during the tornado) in the horizontal and 750 m in the vertical. A
reflectivity threshold of 35 dBZ, as well as 30 dBZ, with a volume of $\geq$16 km$^3$ was chosen for tracking the cells
observed by the BRU radar.

It is important to point out that TITAN not only identifies potentially severe cells, based on a pre-defined reflectivity
threshold and volume, but also tracks them along their life cycle, and additionally provides forecasts of their future
positions up to 60 min. The latter facility assists the meteorologist with the issuing of short-term severe storm warnings
(nowcasting).

Operational Doppler radars do not record the occurrence of a tornado directly, due to its relatively small horizontal
diameter of generally <500 m. However, good indications are being provided by certain signatures in the radial velocity
field, such as regions where opposing radial velocities along the azimuth are observed, which would identify a rotating
circulation associated with a mesocyclone. A tornado will be spawned when the cloud funnel extends to the ground,
where it will create a characteristic damage pattern, which in turn will reveal the magnitude of the phenomenon. The
analysis of the severe storm studied in this paper confirmed the presence of cyclonic rotation (velocity couplet) in the
radar echo core, as well as in the surveyed damage track, which would identify it as a tornadic cell. However, due to the
nocturnal occurrence shortly after midnight (LT), no funnel was reported.

## 3  The event of 04/05 June 2016 in the Campinas region

### 3.1  The Area

Campinas is the third largest town in the State of São Paulo (Fig. 1) and is among the 15 cities with the largest
population in Brazil, comprising >1.2 million inhabitants, with a demographic density of 1,527.6 inhabitants per km$^2$
(IBGE, 2020).

Campinas and neighboring towns have a diverse and dynamic economy. During the last decades the region became one
of the most sophisticated centers for research and development of science, technology and innovation in the country,
concentrating more than 50 branches of the 500 largest companies in the world and accounting for 10% of industrial
production in Brazil in various sectors, such as automotive, textile, metallurgy, food, pharmaceutical, petrochemical,
telecommunications and electronics (Cavalcanti et al., 2017). Campinas is one of the 15 National Brazilian Metropolis
(IBGE, 2018), with 34 cities under its influence; in other words, Campinas is an attraction pole for health services, high
education and technology.

**Figure 1:** (a) South America with the State of São Paulo (black); (b) State of São Paulo with Campinas (black); (c) Campinas with damage track of the event (black); (d) detail of the path of destruction. Information based on field surveys by the authors after the tornado and information from the Campinas Civil Defense Department.

Campinas was the first Brazilian city to be certified as a Resilient City in 2013 (UNDRR, 2020). Created in 2010 by the

UNDRR (United Nations Office for Disaster Risk Reduction), this program aims to contribute to increasing resilience in

local and national contexts, making risk management a component in the development process. However, Campinas



does not have a contingency plan that includes storms with tornadic potential, with pre, during and post event measures, such as alerts on potential occurrences and post disaster measures or shelters to affected people. Furthermore, this international certification that Campinas has held since 2013 could promote a false sense of resilience to the public authorities and the population.

The tornado of 05 June crossed the city from west to east, affecting more than 10 neighborhoods in both the urban and the rural area of the municipality, in a track of approximately 20 km in length and a width of about 1-2 km (Fig. 1d) The affected areas were mostly middle and upper-middle class neighborhoods, with solid constructions, a fact that attests the severity of the phenomenon.

According to the local Civil Defense Department there were no deaths reported, but 1,571 people were affected by the destructive event, of which four were left homeless and four had minor injuries. The Commercial and Industrial Association of Campinas estimates that the damage had exceeded R$ 15.5 million (Digitais PUC-Campinas, 2016) which does not include damage to private homes and suburban infrastructure. An estimated 1,800 trees were uprooted and 500 housing units had suffered some damage. A major shopping mall in the city had been severely damaged. Electricity transmission networks were affected by the winds that knocked down two high voltage transmission towers, leaving about 100,000 residences without electricity for hours. Some remote rural neighborhoods remained without electricity for days. Furthermore, the Civil Defense Department estimated that 300 buildings in a high-income neighborhood were damaged, with an estimated cost of R$12 million for restoration or reconstruction.

According to the Civil Defense Department, 74 mm of rain were recorded in just 45 minutes - an extraordinary value since the monthly average of June for the period 1990-2019 is 42.8 mm, while in June 2016 the total accumulated rainfall recorded was an exceptional 180 mm (CEPAGRI, 2020).

### 3.2 Synoptic Situation

The synoptic situation during the first few days of June 2016 was rather typical for occasionally occurring severe storms during the austral winter months in the State of São Paulo, some of them were reported to have caused considerable damage in the central region of São Paulo state during the period of 04-06 June 2016. On 04 June 2016, a strong anticyclone was centered over the northern half of South America at the 250 hPa level and bounded by a strong zonal Subtropical Jet (STJ) in the south (CPTEC, 2016). The consequential circulation advected moist air from the Amazon and Pacific region throughout the troposphere, resulting in a baroclinic flow with unstable conditions (Fig. 2) favorable for the development of severe thunderstorms in the State of São Paulo, even during the night.

The corresponding GOES13 image in Fig. 3a clearly shows the cloud band within the northwesterly flow around the anticyclone, with several severe storm complexes embedded over Paraguay and the State of São Paulo. Three hours later, Fig. 3b pictures the clouds over Southeast Brazil at 03:00 UT (00:00 LT), the time when the supercell approached Campinas. Cloud tops, with temperatures between -70°C and -80°C, reach up to the tropopause at around 16 km (106 hPa, -76.7°C; UW, 2016). The approximate position of Campinas is indicated.

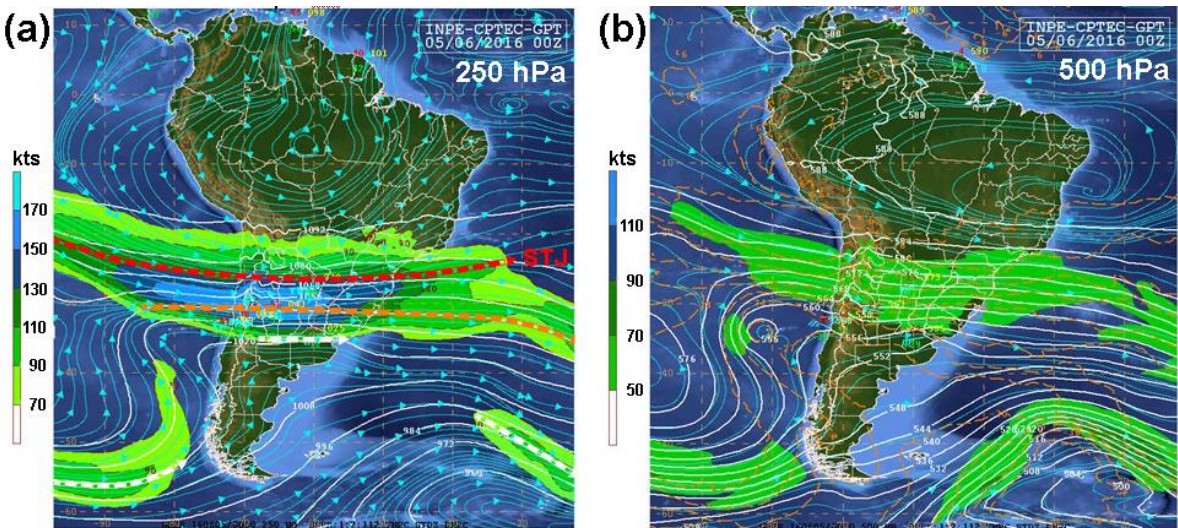

**Figure 2:** Synoptic maps at 00 UT on 05 June 2016: (a) 250 hPa and (b) 500 hPa, adapted from CPTEC (2016).

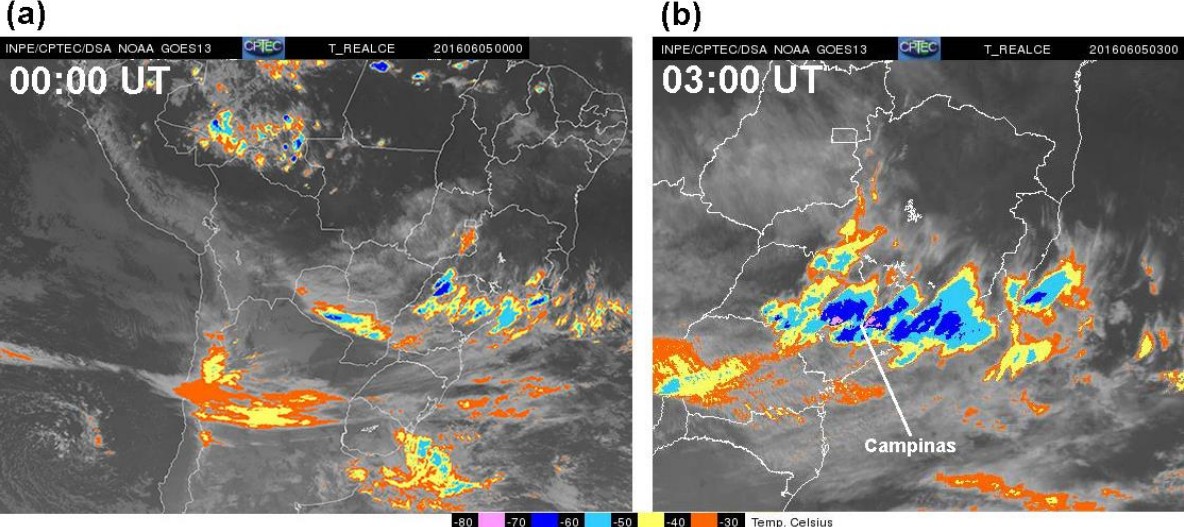

**Figure 3:** Zoomed GOES13 images of T_Realce at (a) 00 UT and (b) 03 UT on 05 June 2016, adapted from INPE/CPTEC/DSA (2016).

## 3.3 Radar Observations

### 3.3.1 Overview of severe convective activity on 04/05 June 2016

The radars in Presidente Prudente (PPR) and Bauru (BRU) operate in Local Time (Brasilia, LT), but TITAN MDV and BrasilDAT data are recorded in Universal Time (LT=UT-3h), as marked appropriately in the figures. Both radars recorded widespread rain south of the Tietê River at 21:00 LT (Fig. 4a) as also depicted in the GOES image at 00:00




UT (Fig. 3a), with embedded cells, driven by the zonal flow in the mid-troposphere, moving eastwards at speeds
ranging from 50–70 km h$^{-1}$.

Figure 4b shows the situation with several severe storm complexes at 00:00 LT, shortly before the tornadic cell
impacted Campinas, coinciding with the GOES image at 03:00 UT (Fig. 3b). The severe convective cells (≥35 dBZ)
correspond well with cloud tops colder than -60°C, as observed by GOES.

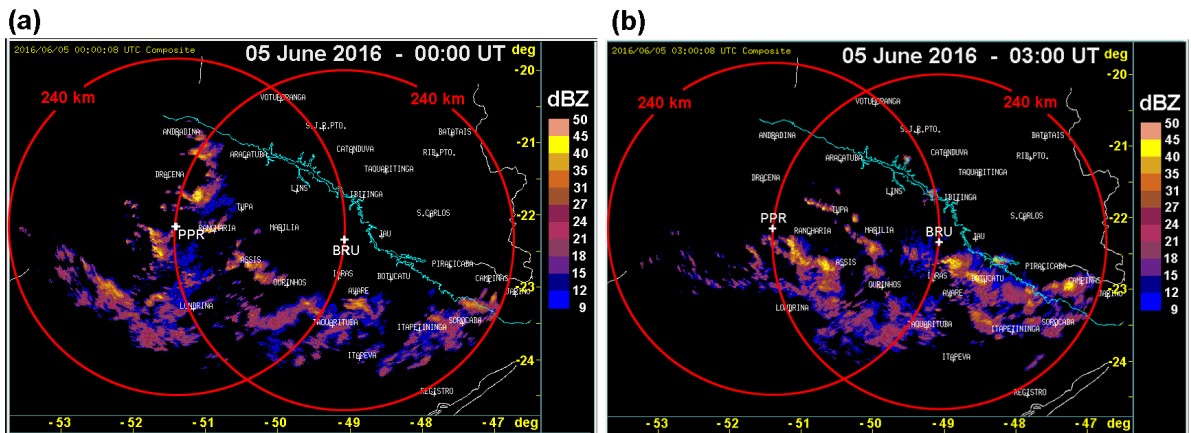


**Figure 4:** Composite TITAN images of the quantitative ranges of the S-band Doppler radars in Presidente Prudente (PPR) and Bauru
(BRU) on 05 June 2016 at (a) 00:00 UT (21:00 LT) and (b) 03:00 UT (00:00 LT).

Figure 5 shows areas along polygons where intense convective activity of ≥35 dBZ reflectivity occurred between 20:21
LT and 00:52 LT when reaching the edge of the quantitative 240 km range, based on TITAN-generated cell tracks.
Each of the beige elements of the polygon corresponds to a Volume Scan at 7.5 min intervals. The last elements of each
track at 03:52 UT are shown in blue. The tornadic cell passed over Campinas at around 00:22 LT (03:22 UT), causing
severe damage at the surface. This cell continued eastwards, also causing damage in Morungaba at around 00:52 LT
(blue area at the end of the track, reaching the limit of the 240 km radar range). The lifetime of many cells during this
event exceeded four hours while traversing the region with storm velocities of >50 km h$^{-1}$, characteristic of severe
storms associated with supercells observed previously in the State of São Paulo (Held et al., 2010a). The cell which
spawned a tornado over the Campinas region can certainly be classified as a supercell, based on several of its
characteristics, such as velocity, echo tops penetrating the tropopause, as well as "severe storm parameters" generated
by the TITAN analysis, which will be discussed later. It should be noted that the BRU radar has a 2° beam width (at 240
km range the aperture already has a diameter of 8 km) and the reflectivity is not range-corrected, resulting in a slight
underestimation of the reflectivity and parameters derived from it by TITAN. In this respect, the SRO radar would have
been in a more favorable position to track the tornadic cell, being only 75 km south of Campinas.



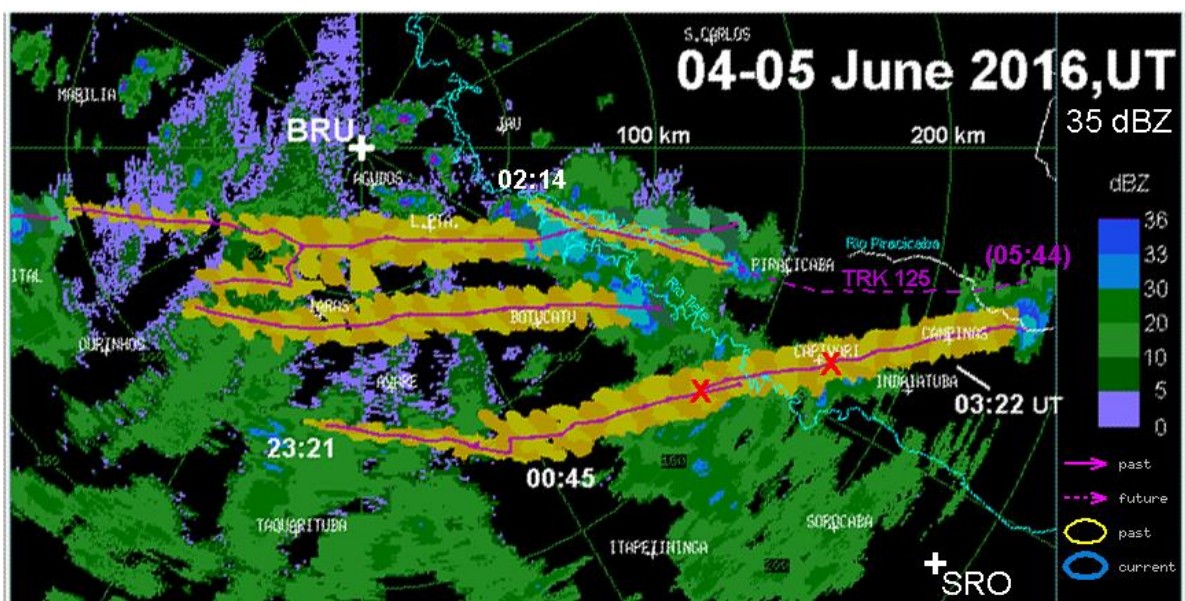

**Figure 5:** Polygons showing the temporal sequence of areas with a reflectivity of ≥35 dBZ, generated with the cell tracking capability of TITAN (Universal Time, LT=UT-3h) until the cell reached the 240 km limit of quantitative observations. BRU marks the origin of the Bauru Doppler radar and SRO the location of the Radar at São Roque. Cell positions at 03:52 UT are shown in blue. The red x marks positions of earlier strong convective activity (01:37 and 02:22 UT, respectively).

In order to support the finding that the tornado had been spawned by a supercell, raw data (HDF5 format) from the S-band radar at São Roque (SRO) had also been converted to MDV, allowing the generation of cell tracks beyond the 240 km range of the BRU radar, as shown in Fig. 6. Due to the fact that the BRU and SRO radars are not integrated, because they are being operated by separate organizations, as well as having slightly different characteristics, one has to expect some differences in the images. For this reason, TITAN cell tracking was performed with a reflectivity threshold of 40 dBZ. It can be concluded from Fig. 6 that the supercell spawning the tornado over Campinas had a lifetime of ca 8.5 hours from its first echo (FE) at 22:59 UT until it finally dissipated 205 km northeast of the SRO radar at 07:30 UT, although the actual tracking of the 40 dBZ echo core stopped at 06:50 UT.

Furthermore, Fig. 6 highlights the complex convective situation during this event, resulting in several long-lasting cells being active in the region, but at differentiated times. One of these originated with its FE at 21:07 UT near the town of Taquarituba (FE not shown in Fig. 6), initially following the same track as the supercell, which was later tracked traversing Campinas. It rapidly grew in volume while moving towards east-northeast (blue area at 23:20 UT north of Sorocaba, Fig. 6), later turning into a supercell. At around 00:30 UT, a new cell had suddenly developed on its right flank, possibly in the downdraught region of the mother cell, subsequently splitting off and moving into the Paraiba Valley (south of the Mantiqueira Mountain Range; passing over São José dos Campos). The original cell continued on its initial track east-northeastwards, causing severe damage in the small town of Itupeva (at around 00:30 UT), before reaching Jarinu and Atibaia (00:50-01:19 UT), about 40 and 50 km, respectively, southeast of Campinas (Fig. 6). While this supercell now moved at ±60 km h$^{-1}$ towards east-northeast, it increased its VIL by >160% during 40 min until reaching a peak at 00:50 UT. This would imply a very strong updraught which was probably responsible for spawning





an EF3 tornado in Jarinu between 00:50-01:10 UT, as reported by eyewitnesses and confirmed by a subsequent ground survey, revealing enormous damage in the town, including two overturned semi-truck trailers, as well as causing one

fatality. Due to the lack of radial velocity data, no rotation inside the cell could be documented. Thereafter, this supercell continued parallel to the one that had split off at 00:30 UT (Fig. 6), but on the northern flank of the mountain range, into Minas Gerais state. Both supercells dissipated shortly outside the 250 km range of the SRO radar at around 04:00 UT, having had an overall lifetime of almost 6 hours. No further significant damage had been reported from either supercell.

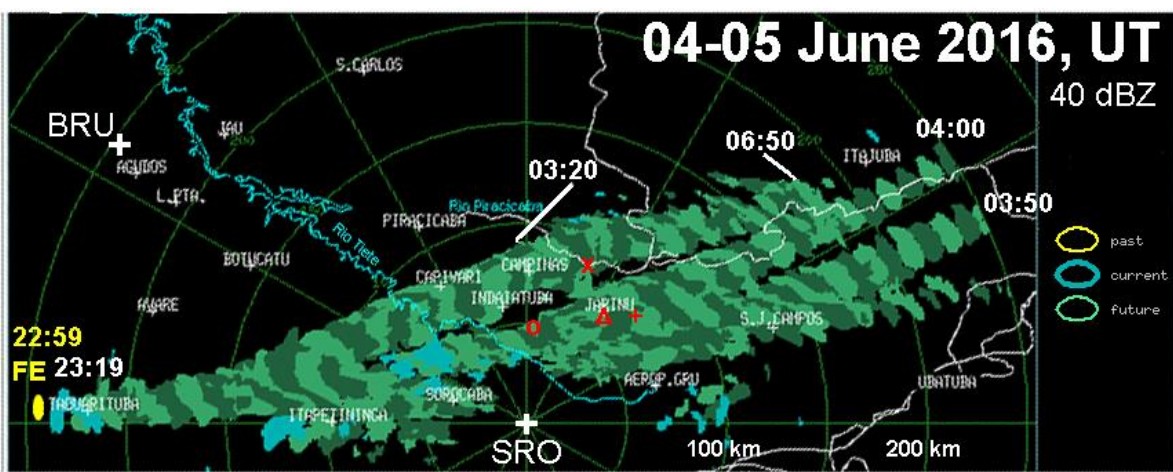


**Figure 6:** Polygons showing the temporal sequence of areas with a reflectivity of ≥40 dBZ, generated with the cell tracking capability of TITAN as observed by the Radar at São Roque (SRO) from 23:00-04:00 UT. BRU marks the location of the Bauru radar. Blue elements show simultaneous complexes at 23:19/23:20 UT with forward projection at 10 min intervals in green. Red symbols mark the following towns: x Morungaba, o Itupeva, Δ Jarinu, + Atibaia.


Another long-lasting cell first detected (≥35 dBZ) ca 60 km east-southeast of BRU at 02:14 UT (Fig. 5), initially moving along the Tietê River towards southeast, but later in a more easterly direction, passing near Piracicaba, until reaching the limit of the 240 km range at 05:44 UT (TRK 125; Figs 5 and 11). Its life span was >3.5 hours. Due to various tracks overlapping each other, Figs 5 and 6 were configured in such a way as to only show relevant tracks

during the time interval indicated.

All these major storms were typical multi-cellular complexes, propagating erratically at speeds varying between 50 and 70 km h$^{-1}$, with maximum reflectivities between 55 and 65 dBZ during their mature stages, occasionally overshooting into the lower stratosphere. Fluctuating speeds and reflectivity values along the cell track are indicative of rapid development of new cells ahead or laterally in the outflow region of the mother cell. However, in this paper,

characteristics only of the tornadic cell passing over Campinas will be discussed in more detail based on data of the BRU radar.





### 3.3.2 Details of the Campinas supercell and tornado

Figure 7a shows a zoom of the cell track above the Campinas region. The light-blue area of the polygon identifies the cell at 00:22 LT (03:22 UT), which is more or less the time when the tornado had reached the ground, while yellow and

beige areas indicate the storm position every 7.5 min before, and green areas mark its future track until reaching the 240 km range from the radar. Furthermore, the cross-section in Fig. 7b also provides details of the vertical structure of the cell along the base line A-B, oriented perpendicular to the cell motion, over the urban region of Campinas, with a reflectivity maximum of 46 dBZ extending to >6 km of altitude and echo tops reaching ≤18 km (the 10 dBZ reflectivity contour has been extrapolated, based on overshooting cloud top temperatures; Fig. 3b). These parameters indicate

extremely strong convective activity, considering the fact that this storm occurred during the generally dry month of June. Coincidental with the area of maximum reflectivity, a typical characteristic of tornadic cells can also be observed in Fig. 7c, viz., an area of maximum radial velocities away from the radar (positive velocities; warm colors) opposing in azimuth radial velocities towards the radar (negative velocities; cold colors). This pair of opposite radial velocities ("*couplet*") identifies the presence of a mesocyclone, indicative of the existence of rotation within the storm, which can

spawn a tornado.

The rotational velocity is defined as modulus of the radial velocities divided by 2, viz., $(|V_{in}| + |V_{out}|)/2$, which then permits the calculation of the relative rotational velocity of the tornadic cell affecting Campinas during this event. The radial velocities found in this analysis ranged between -25 m s$^{-1}$ (towards the radar) and +15 m s$^{-1}$ (away from the radar). Based on these radial velocities recorded by the Doppler radar in Bauru, a value of 12.5 m s$^{-1}$ is derived for the

rotational velocity $V_r$, where the centers of maximum radial velocity were separated by a distance of 5 km. It should be pointed out that this velocity couplet was observed despite its radial distance of just over 200 km from the origin of the radar in Bauru. Considering the beam width of the Bauru radar, a significant degradation of the vortex at that distance should be expected (Brown and Wood, 1999). The rotational vorticity can be calculated from the ratio between the speed and the distance between the pair of opposing radial velocities, yielding a value of 2.5·10$^{-3}$ s$^{-1}$ for this case. From

studies of tornadic storms in the USA, threshold values of $V_r \geq 12.5$ m s$^{-1}$ within a radius of 150km, and ≥8.5 m s$^{-1}$ for distances further than 150km were defined (NSSL, 1985). Thus, the values associated with the Campinas cell can be considered to characterize an intense tornadic event. The harmful consequences associated with the event, such as the uprooting of many dozens of large trees, total or partial destruction of a large number of residences and impacts on the urban infrastructure also attest the severity of the phenomenon.

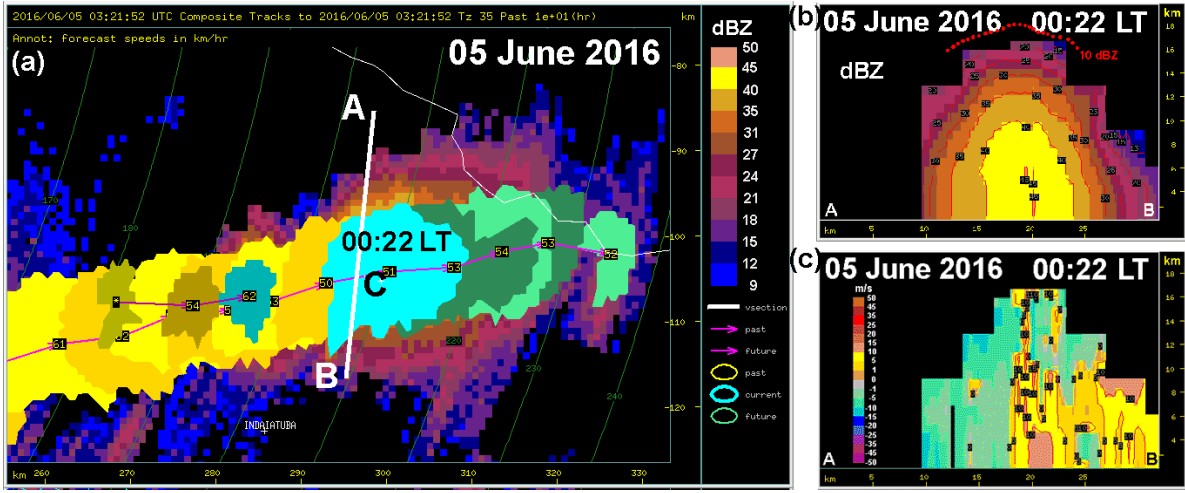

**Figure 7: (a)** Polygons representing areas of reflectivity ≥35 dBZ identified and tracked using the software TITAN. **(b)** Vertical cross-sections along the line A-B show the vertical extent and reflectivity structure and **(c)** radial velocities associated with the tornadic cell impacting Campinas at 00:22 LT

Time sequences of a variety of severe storm parameters were calculated by TITAN for all cell tracks shown in Fig. 5 (BRU) and Fig. 6 (SRO), from which the following were considered from the supercell that passed over Campinas and are graphically presented for BRU in Fig. 8: Echo Top (10 dBZ contour); Max Reflectivity (dBZ); VIL (kg m$^{-2}$) and propagation velocity of the 35 dBZ echo cores. In addition, Fig. 8 also shows the Total Lightning (number of strokes per 7.5 min Volume Scan within the 30 dBZ reflectivity contour). Although most parameters compare well between

BRU and SRO during the common time period (from the FE at 23:19 UT until 03:52 UT), but maximum reflectivities from SRO were around 15 dBZ higher and consequently yielding unrealistically high VIL values. Therefore, only track data from BRU will be presented in this study, despite the great radial distance of the tornadic cell during its most intense stage, resulting in an underestimation of especially maximum reflectivities and their heights (volumes) and subsequently VIL data, as explained above.

From eyewitness reports it is concluded that the tornado occurred during the 03:14–03:29 UT Volume Scans, with maximum intensity at around 03:22 UT. As can be seen from Fig. 8, the maximum reflectivity and VIL reached a peak during the 02:59 UT Volume Scan, well before the tornado had spawned and reached the ground. Their observed values of 49.5 dBZ and 15.8 kg m$^{-2}$, respectively, are certainly an underestimate, but considerably lower than those recorded during the Indaiatuba EF3 tornado (about 25 km south-southwest of Campinas, also at a range of ≥200 km from BRU),

viz., 57.5 dBZ and 40.5 kg m$^{-2}$, respectively (Held et al., 2010a; Nascimento et al., 2014). Although both cases must be regarded as underestimates due to the large distance of >200 km, there may also be other reasons to be considered. The tornado in Indaiatuba was certainly more





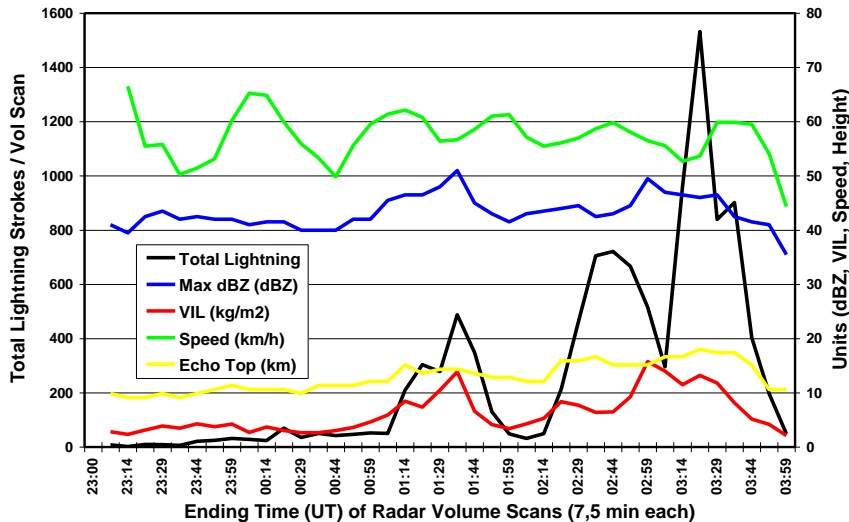

**Figure 8:** Time sequence of TITAN-generated storm parameters per Volume Scan of the Bauru Radar (Campinas track; 04 June 2016, 23:00 UT to 05 June 2016, 04:00 UT): Reflectivity maximum (dBZ), VIL (kg m$^{-2}$), Speed of Cell Propagation (km h$^{-1}$) and Top of 10 dBZ reflectivity contour (km amsl), compared to the number of CG-negative strokes (black line) per Volume Scan.

severe, based on the damage, but also it occurred during the meteorological transition phase of austral autumn, while the current case happened during the dry winter season, when convective cells are less intense. Figure 8 also shows that the speed with which the tornadic cell propagated throughout its lifetime varied between 50-65 km h$^{-1}$, which is characteristic for rapid new cell development in the immediate vicinity, and especially ahead of the mother cell. This is also related to peaks in VIL and lightning stroke frequency during the earlier life cycle of the supercell, at around 01:37 UT and 02:22 UT, identified in Fig. 5 with x. During the first two hours of the super cell's lifetime the echo tops (10 dBZ) fluctuated around 10 km, but began to rise after 01:00 UT as the storm intensified, occasionally penetrating the tropopause (at 16 km), also indicated by the other parameters shown in Fig. 8. The overshooting echo top reached a maximum of 18 km, simultaneously with a well-pronounced peak in lightning activity. While the tornadic cell passed over Campinas, TITAN indicated a "Probability of Hail" (PoH) of 60-80%, but FOKR (Foote Krauss) category 2 (of 4) only. The PoH is in accordance with ground observations during the first minutes of 05 June 2016 (LT), when the storm impacted the urban area of Campinas. During this period, the "HailMassAloft" reached a maximum of 10.8 ktons. Hail falling on the ground was reported by residents around this time.

It is noteworthy that the Campinas tornado was spawned directly from the mother cell, while all earlier case studies (Held et al., 2010a) showed the tornado-spawning cell to be an isolated cell trailing the storm complex.

### 3.4 Damage

A field recognition conducted along the storm path shortly after the occurrence of the event in Campinas revealed typical signatures of an EF2-3 tornado. It is noteworthy that the affected neighborhoods are middle to upper-middle class with houses having brick and cement walls, which are considerably more resistant than the American frame





homes. Therefore, the degree of destruction in Campinas highlights both the destructive power of the phenomenon and the difficulty of adapting the EF scale to the Brazilian case. The photos in Fig. 9 document some of the damage caused by the storm.

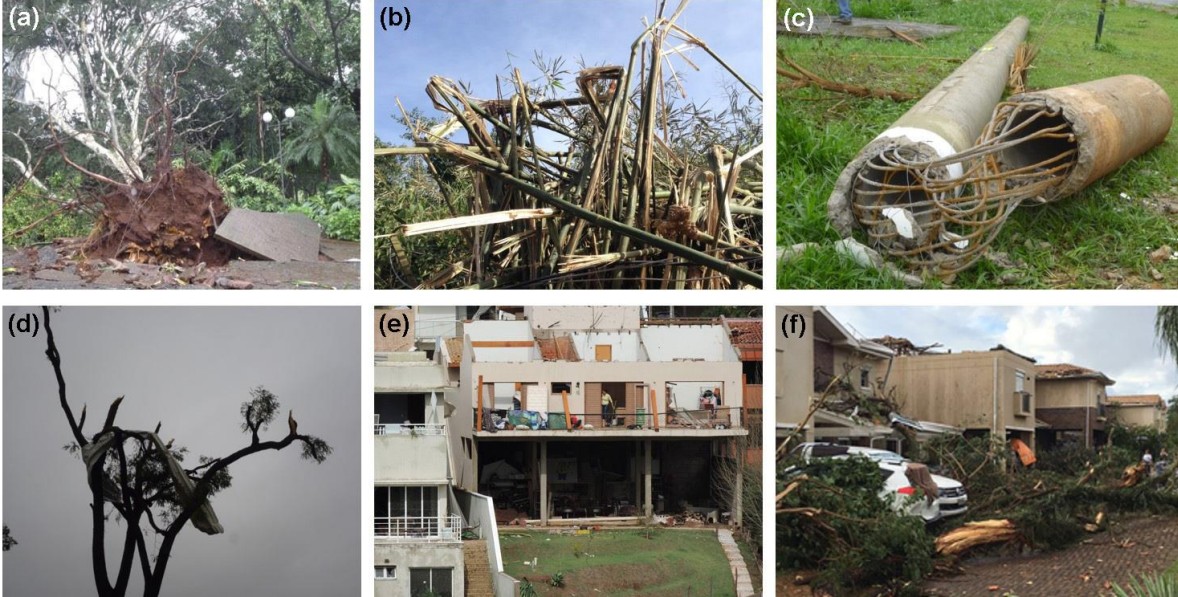

**Figure 9:** Damages associated with the tornado passage in Campinas: (a) tree uprooted; (b) twisted branches demonstrating tornado rotation; (c) totally destroyed electric pole; (d) large airborne pieces; (e) and (f) major damage to homes and roofs. Credits: Lucí H. Nunes, Sonia Tikian and Civil Defense of Campinas.


There are no photographic records or videos of the probable tornado funnel available, most likely due to the nocturnal occurrence of the storm, as well as the power outage, which made it impossible for surveillance cameras scattered around various parts of the city to provide any kind of registration. Residents of the affected areas reported a loud noise similar to an explosion or to an airplane turbine.

About three hours before the tornado in Campinas, severe wind damage occurred in Jarinu, a small town about 40 km
southeast of Campinas. It can be attributed to an earlier supercell traversing the region (Fig. 6). An on-site survey and informal interviews with residents and civil defense officials leave no doubt that a severe event took place in the town between 21:50-22:19 LT on 04 June. Although this area is outside the Bauru radar coverage, the evidence strongly suggests that it also was a tornadic cell, as can be seen in Fig. 10, with metal structures twisted around light poles, concrete poles twisted, large trees uprooted, as well as widespread damage to homes and commercial buildings. The
local press reported public and private losses above R\$ 18 million, with 275 homes damaged and 83 totally destroyed, two overturned semi-truck trailers, as well as injured people and one fatality. Eyewitnesses also mentioned that at least three cars were thrown several meters away. Based on the massive destruction, the phenomenon could be rated as an EF3 tornado.



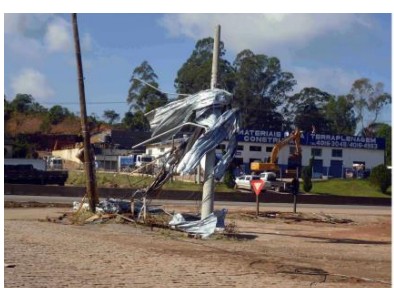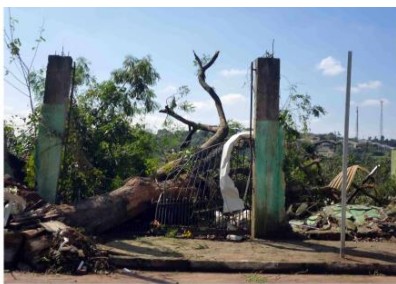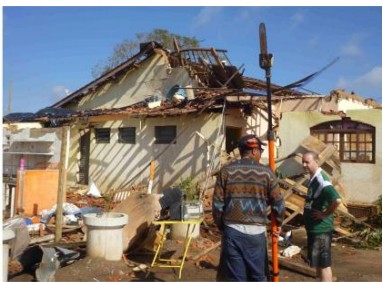

**Figure 10:** Damage caused by the supercell in the town of Jarinu on 04 June 2016. Credits: Lucí H. Nunes and Raul R. Amorim.

Other nearby towns registered severe weather conditions, as reported by local medias. A severe event occurred in the town of São Roque, where the SRO radar is located, during the afternoon of 06 June 2016, causing considerable damage, displacement of around 50 people, injures and one casualty, which might have been provoked by a tornado
(Almeida and Lombardi, 2019).

### 3.5 Lightning Observations

Lightning strokes recorded by BrasilDAT (Naccarato et al., 2012; Naccarato and Pinto, 2012) provided the data utilized in the subsequent analysis. The BrasilDAT is a new lightning detection network based on the EarthNetworks technology, with its first sensors having been deployed in Brazil in 2010. It was gradually expanded and reached an
optimal coverage of the State of São Paulo in 2012. Subsequently, the expansion program continued and now the network comprises 65 sensors covering 10 states at southeast, south, center and part of northeast of Brazil.

The TITAN storm tracking software was run with reflectivity thresholds of 30 and 35 dBZ to generate ellipses for each 7.5 min volume scan. Figure 11a shows these ellipses for the 30 dBZ threshold from 23:00 UT (20:00 LT) until 04:00 UT (01:00 LT). Figure 11b highlights the vector track of the supercell (marked as "TRK 5/"), which finally spawned the
tornado in Campinas, from its origin at 23:06 UT until it reached the 240 km quantitative range of the Bauru radar (BRU) as TRK 5/73. These tracks are based on the center point of the ellipsoids around the 30 dBZ reflectivity contour (Fig. 11a). Since TITAN allows merging or splitting of cells, the sub-number of the complex (TRK 5/…) changes along its path, as can be seen in Fig. 11b. The ellipses from the tornadic cell "TRK 5/" were consolidated into a continuous polygon, which delineated the area for which stroke data were extracted for the period 23:00 UT until 04:00 UT.
Subsequently, they were separated into positive and negative cloud to ground (CG) strokes, as well as intra-cloud stokes (IC) for further analysis.

A first glance at the lightning stroke data reveals an exceptionally large number of negative CG and IC strokes within the 30 dBZ polygon, viz., 2145 negative CG strokes and 8689 discharge events classified as IC, during the period 23:00-04:00 UT. It is also noteworthy, that 52.5% of the ground strokes recorded within the 30 dBZ contour were
confined within the 35 dBZ envelope, while 58.1% of IC activity occurred within the 35 dBZ contour. Maximum Peak Currents recorded for CG-neg and CG-pos strokes were -131 kA and +133 kA, respectively, with the most frequent intervals being 10-19 kA and 20-29 kA, respectively. However, only 0.8% of the CG-neg strokes were ≥50 kA. The





extreme peak currents are actually isolated outliers, which occurred during the most intense phase of the supercell (03:17-03:18 UT), which is coincident with the tornado having impacted on the ground.

**(a)**

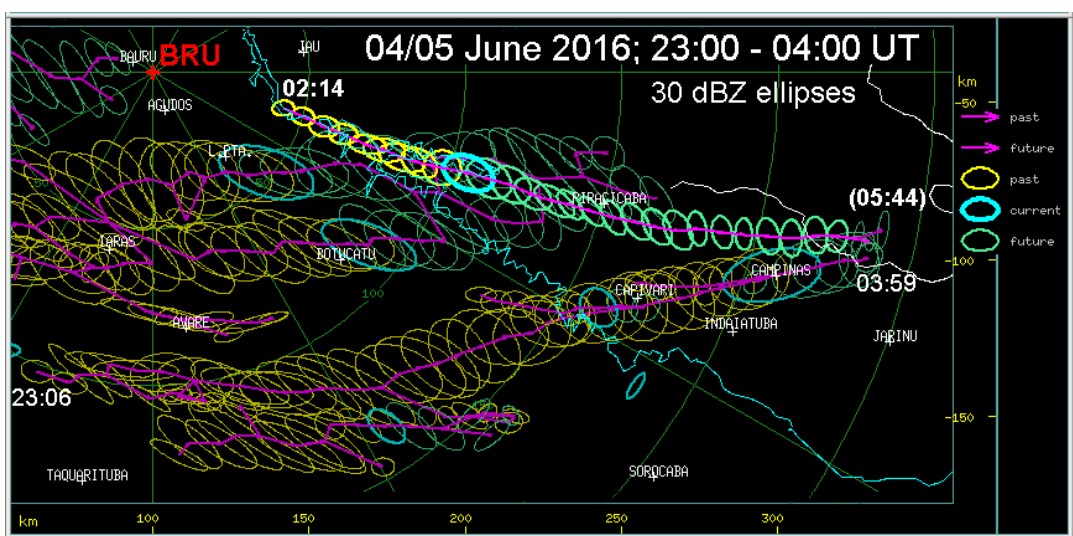

**(b)**

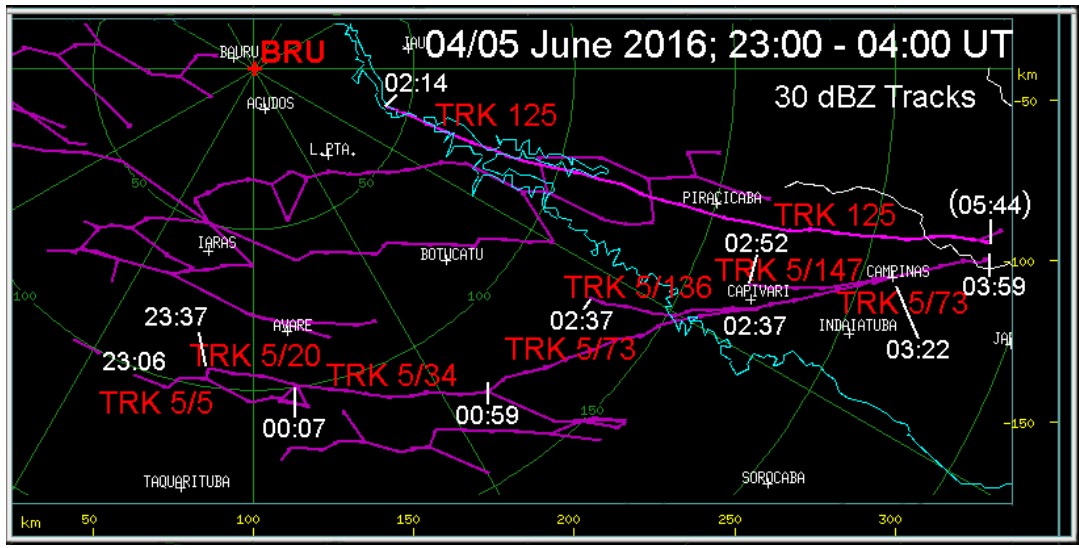

**Figure 11: TITAN-generated cell tracks from 20:00 LT on 04 June 2016 until 01:00 LT on 05 June 2016 for a reflectivity**
**threshold of 30 dBZ. (a) Ellipses for every volume scan; (b) Track vectors, also showing merging and splitting of cells. BRU marks the origin of the Bauru Doppler radar**

The time sequence of these lightning strokes per minute is presented in Fig. 12. Three distinct peaks can be seen, of which the first two are associated with rapid cell development and marked with x in Fig. 5. The third peak evidences a

typical *lightning jump*, starting just after 03:00 UT and reaching a maximum of 238 strokes min$^{-1}$ (total lightning) 14





minutes later, preceding the assumed touch-down time of the tornado (estimated to have occurred at around 03:22 UT) in the urban area of Campinas. The green arrows mark the peak of the *IC lightning jump* with 185 strokes min[-1] at 03:14 UT, followed by a sharp drop to 72 strokes min[-1] at 03:25 UT, just 3 minutes after the tornado touch-down estimated time. After about five minutes, the IC flash rate enhanced very quickly again, reaching over 110 IC strokes min[-1], in

accordance with the studies of MacGorman et al. (1989) for the Binger storm.

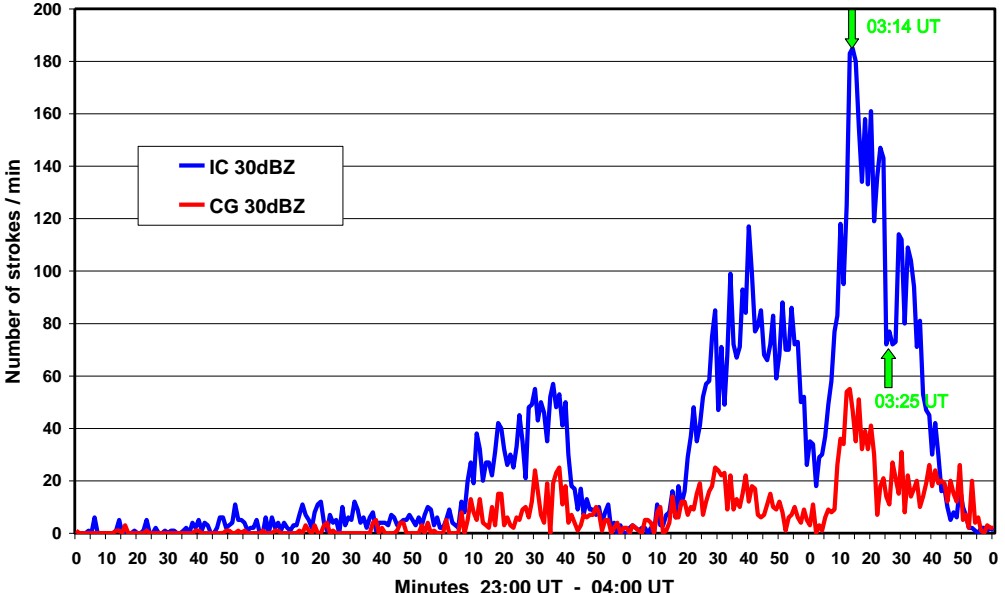

**Figure 12: Number of CG-neg strokes and IC events per minute within the 30 dBZ reflectivity contour of the tornadic cell on 04-05 June 2016, 23:00 UT-04:00 UT.**

The large number of strokes produced by the event (Fig. 12) has to be highlighted, particularly the *lightning jump* effect, which is a very rapid intensification of the electrical activity inside the thundercloud (Goodman et al., 1988, 2005; Williams, 1989; Gatlin and Goodman, 2010; Schultz et al., 2011). These authors presented studies in which storms demonstrated a sudden increase in lightning activity just before the occurrence of a severe event, such as hail, tornado, wind gusts, and downbursts or microbursts. This relation of the electrical activity with the severe weather occurrence

can be explained by the relationship between the strong updrafts within the cloud and the electrification of the hydrometeors. The updrafts together with the gravitational forces distribute the hydrometeors within the cloud according to their size, which are then quickly electrified by non-inductive electrification processes. The result is a fast formation of positive and negative charge centers within the cloud that leads to an enhancement of the lightning production (Deierling et al., 2008). In general, the intensification speed of the lightning jump phenomena (curve slope)

is directly related to the increase of the updrafts (or convection invigoration), which produces much more ice within the cloud, but with smaller size and density, leading to a higher number of collisions, higher charge transfers and consequently more intense and larger charge centers.





Figures 13 and 14 depict the tornadic cell at 03:07 UT (00:07 LT) while the lightning frequency was at a low (33 CG-neg strokes per7.5min), just before the *lightning jump* and at 03:22 UT (00:22 LT) while it traversed densely populated
suburbs of the city of Campinas. These times were selected based on Severity Parameters calculated by TITAN, as well as witness reports, which identified this period as the most critical of the event. Each of the volume scans has a duration of 7.5 min and all negative CG lightning strokes, represented by a yellow + during the corresponding time period, are superimposed on a CAPPI reflectivity image. The exceptionally high number of strokes, with a frequency of up to 301 CG-neg strokes per 7.5min during the 03:14-03:22 UT Radar Volume Scan (Fig. 8), includes the maximum of the
lightning jump in Fig. 12 (185 CG-neg strokes per minute at 03:14 UT). It is also noteworthy, that from 03:17 UT to 03:24 UT only 3 positive CG strokes were recorded, while thereafter until 03:40 UT no positive strokes were observed.

Figure 13 clearly shows that the stroke count for the 02:59-03:07 UT Volume Scan originates from three separate cells, but only seven strokes were recorded ahead of the tornadic cell core, which is in agreement with observations from previous case studies, where mostly an absence of CG strokes was observed during the tornadic events (the available
Lightning Network, RINDAT, at that time could not discriminate IC strokes at that time). However, during the assumed touch-down of the tornado in the current case, intense lightning activity of both CG and IC strokes occurred, as shown in Fig. 12 and Fig. 13, 03:22 UT. At this stage, no definitive reasons for this behavior can be suggested, but it might be attributed to the fact that this tornadic storm was the first of its kind documented by radar during the dry winter season.

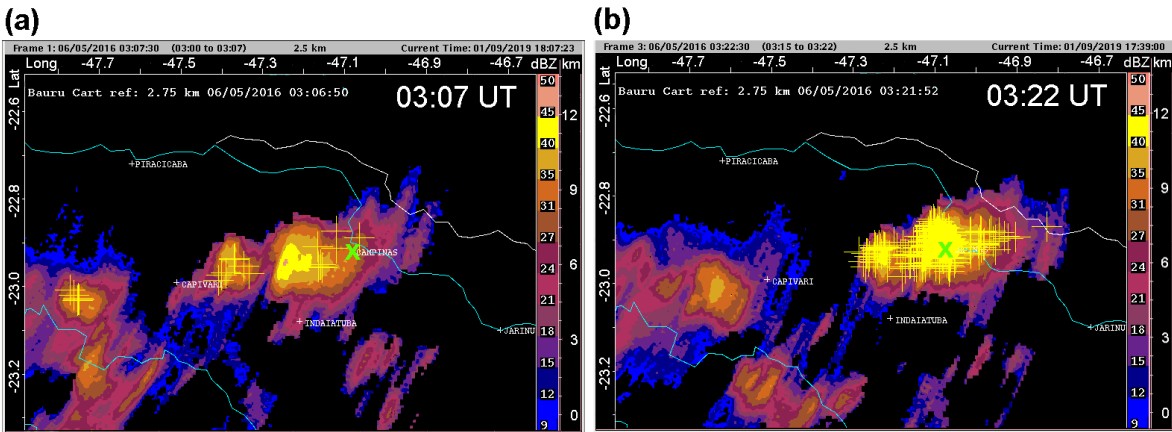

**Figure 13: Tornadic cell approaching / traversing Campinas on 05 June 2016, at (a) 03:07 UT and (b) 03:22 UT, respectively; reflectivity field (CAPPI at 2.75 km amsl) with superimposed negative CG lightning strokes. The green "X" marks the center of Campinas.**

Vertical cross-sections for the above Volume Scans are shown in Fig. 14. The respective base lines are indicated in the
zoomed CAPPIs of the principal tornadic cell, and the different electrical behavior before and during the tornado touch-down is shown at the base of the cross-sections in form of red lines, representing regions with CG activity. The characteristic hook echo in the CAPPI of 03:22 UT marks the low-level inflow region into the mesocyclone at the time the tornado was spawned. This also coincides with the sharp lightning drop (Fig. 12). The dotted line indicates the





position of the cross-section showing radial velocities in Fig. 7, while the blue and red arrows represent the opposite
radial air flow ("couplet") powering the mesocyclone within the supercell.

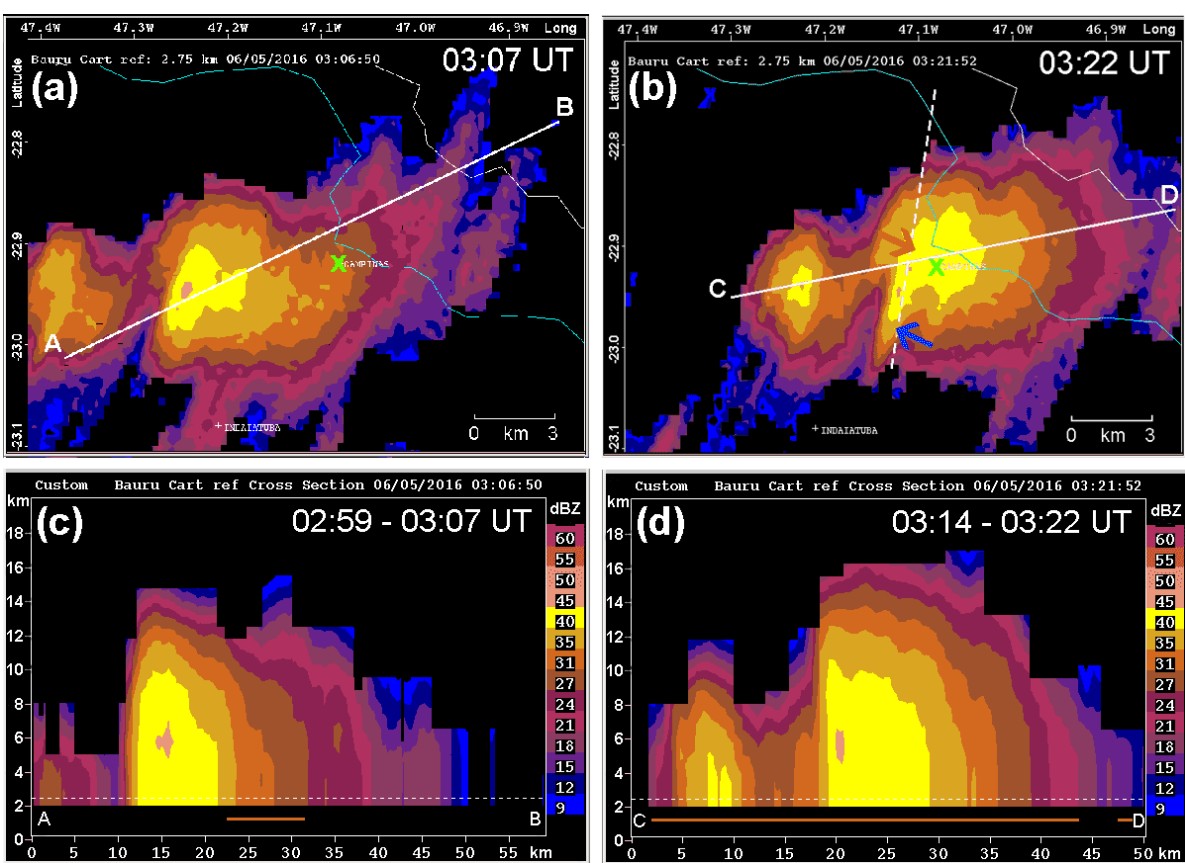

**Figure 14: Zoom of the tornadic cell on 05 June 2016, (a) 03:07 UT and (b) 03:22 UT; CAPPI at 2.75 km, reflectivity field with the respective base line of the vertical cross-sections. "X" marks the center of Campinas. The dotted base line in the**
**CAPPI 03:22 UT indicates the position of the vertical cross-section in Fig. 7 and the short arrows represent the radial air flow towards (blue) and away (red) from the radar, characterizing the mesocyclone; (c) and (d) Vertical cross-sections with ground positions of CG-neg strokes (red horizontal lines).**

## 4 Conclusions

The synoptic situation during the first few days of June 2016 was rather typical for occasionally occurring severe storms
during the dry austral winter period in the State of São Paulo. A strong anticyclone was centered over the northern half
of South America at the 250 hPa level and bounded by a strong zonal Subtropical Jet (STJ) in the south. Thus, moist air
was advected from the Amazon and Pacific region throughout the troposphere, resulting in a baroclinic flow with
unstable conditions favorable for the development of severe thunderstorms in the State of São Paulo, even during the
night.

A supercell storm lasting 8.5 hours traversed the eastern half of the State of São Paulo during the night of 04/05 June
2016 and spawned a tornado in the City of Campinas during the early hours of the morning (Local Time). Observations



from three meteorological Doppler S-band adars in the State of São Paulo (Presidente Prudente in the west, Bauru in the central State and São Roque, SRO, 75 km south of Campinas and 245 km southeast of Bauru) were used to track and document the events during this night. Their propriety raw data were converted to MDV (Meteorological Data

Volume) format and processed with NCAR's (National Center for Atmospheric Research) software "TITAN" (Thunderstorm Identification, Tracking, Analysis and Nowcasting) with a spatial resolution of 750 m horizontally and vertically. Despite its distance of ≥200 km from the tornado, the Bauru radar (BRU) was chosen for the detailed analysis due to a better data quality than the nearby SRO radar. As can be seen from the TITAN-generated cell tracks, using a reflectivity threshold of 35 dBZ with a minimum volume of ≥16 km$^3$, several severe long-lasting multi-cellular

complexes were active during the night, propagating erratically at velocities varying between 50 and 70 km.h$^{-1}$, with maximum reflectivities during their mature stages between 55 and 65 dBZ, occasionally overshooting into the lower stratosphere. A variety of important parameters, such as Area, Volume, Precipitation Flux, VIL (Vertically Integrated Liquid water content), Maximum Reflectivity, Hail Metrics, speed and direction of propagation, etc, per volume scan, were calculated throughout the lifetime of storms. Some of these values were found to be lower than observed from

previous supercell and tornado cases, e.g., maximum reflectivity = 49.5 dBZ and VIL = 15.8 kg m$^{-2}$, respectively, which are certainly an underestimate due to the long radial distance, but possibly also because this was the first tornadic event ever recorded by radar during the austral dry winter period and furthermore, during the night, when convective activity is less intense than during daytime. The peaks in radar reflectivity and VIL aloft occurred shortly before the onset of the most intense phase of the storm, accompanied by a heavy downpour of rain and hail, while the lightning activity

dropped to a minimum, before a significant *"lightning jump"* was recorded. Other typical tornado signatures, such as a rotational damage pattern of uprooted and broken trees, as well as a hook echo and a mesocyclone with a rotational velocity of 12.5 m s$^{-1}$, both observed in the radar data at the time of the assumed spawning of the tornado (Volume Scan 03:14-03:22 UT). A vertical cross-section through this Volume Scan clearly pictured a pair of opposite radial velocities (*"couplet"*) at a distance of 5 km, which identifies the presence of a mesocyclone, indicative of the existence of rotation

within the storm, which can spawn a tornado.

Lightning stroke data obtained with the Brazilian Lightning Detection Network (*Rede Brasileira de Detecção de Descargas Atmosféricas Totais,* BrasilDAT), were analyzed in order to characterize the electric properties of the supercell throughout its 8.5-hour lifetime. A first glance at the lightning stroke data reveals an exceptionally large number of negative CG (Cloud to Ground) and IC (Intra-Cloud) strokes within the 30 dBZ polygon, viz., 2145 negative

CG strokes and 8689 discharge events classified as IC, during the period 23:00-04:00 UT (96% of total lightning occurred during the period from 01:00-04:00 UT). Besides two earlier distinct peaks in lightning frequency, associated with rapid cell development, the third peak evidences a typical *lightning jump*, starting just after 03:00 UT and reaching a maximum of 238 strokes min$^{-1}$ (total lightning) 12 minutes later, preceding the estimated touch-down time of the tornado (around 03:21 UT) in Campinas. This is also supported by the observed sharp drop of the lightning frequency at

03:25 UT (from 185 to 72 IC strokes min$^{-1}$), followed by a very rapid increase again after about five minutes. During earlier studies of supercells and tornadoes in the State of São Paulo, an absence of lightning strokes to ground (CG; the then available lightning data did not include IC activity) during the tornado touch down, was documented. In the current



case it was also found that very few CG strokes were recorded shortly before the *lightning jump*. In fact, during the Volume Scan 02:59-03:07 UT only seven CG-neg strokes were recorded ahead of the tornadic echo core (≥40 dBZ), as

revealed from CAPPIs with overlaid strokes, generated by TITAN/CIDD, but thereafter the frequency increased rapidly until 03:14-03:22 UT with up to 55 CG strokes per minute striking the ground anywhere within the 30 dBZ reflectivity contour.

According to eyewitnesses, another tornado spawned in the same region from a different supercell some three hours before the one that impacted Campinas, traveling on a parallel track. Based on the damage pattern in the small town of

Jarinu, it could be considered as an EF3 tornado.

Considering the relatively rare occurrence, small-scale features and short duration of tornadic cells in the central region of the State of São Paulo, it is almost impossible to predict a possible formation of tornadoes timely for issuing warnings to the population. Not even Nowcasts based on continuous radar monitoring would facilitate experienced radar meteorologists to emit a tornado alert, because supercells in this region are generally difficult to identify during

their early stages.

The occurrence of the phenomenon described in this paper has clearly shown that the municipal government and citizens are relatively unprepared to face this kind of severe event, not only in the Campinas region, but also in other parts of Brazil where tornadoes occasionally occur. Furthermore, the currently available Doppler radars in Brazil are insufficient for a countrywide coverage to detect severe storms in real time for issuing timely warnings. The fact that

Campinas was the first Brazilian city certified as a resilient city, crowning its efforts to fight floods, could give a false impression of resilience to other phenomena that cause great impacts, such as tornadoes or devastating hailstorms, for which Campinas and other Brazilian cities are not prepared.

As there is no way to eliminate or even prevent damages caused by a tornado, measures need to be taken to minimize the impact and especially, to reduce the number of injuries and to prevent deaths. This would require an all-out effort to

educate the population countrywide how to react in such an emergency at short notice, like evacuation of precarious buildings, sheltering in a safe room, avoid being near trees and power lines, etc. It should be acknowledged that the local Civil Defense Authorities have already initiated such activities, but it will probably take a whole generation before having reached a major portion of the population in order to become routine.

However, a positive highlight was that post-event measures, such as removal of debris and fallen trees, road clearance

and the re-establishment of energy were well organized, prompt and effective. This is probably due to the fact that Campinas and even Jarinu routinely experiences flooding, so that there is a preparedness of authorities for post-disaster emergency action. The big challenge, however, is pre-disaster warning and the preparedness of people how to prevent further damage and injuries.



*Author contributions.*   LHN and RRA conducted the damage analysis. GH contributed the synoptic analysis and together with AMG analyzed the radar data. GH and KPN processed the lightning data. All authors prepared their relevant sections.

*Competing interests.*   The authors declare no competing interests.

*Acknowledgements*

The authors would like to acknowledge the residents of the affected area who gave their testimony, Mrs. Sonia Tikian for providing photos and information, as well as Prof. Dr. Jansle Vieira Rocha (FEAGRI / UNICAMP) for accompanying the fieldwork in affected rural sectors of Campinas. The Civil Defense Department of Campinas is thanked for collaboration and sharing their information.
The *Centro de Meteorologia de Bauru* (CMB/UNESP) is thanked for providing the Bauru and Presidente Prudente
radar data in IRIS format, as well as for assisting with the conversion to MDV with 250 and 750m range bin resolution (special mention is due to Jaqueline M Kokitsu and Hermes A G França). The *Departamento de Controle do Espaço Aéreo* (DECEA) and the *Centro de Previsão de Tempo e Estudos Climáticos* (CPTEC/INPE) are acknowledged for permitting the use of raw data (HDF5 format) from the São Roque radar, which were kindly converted to MDV by Rafael Vernini (MSc student at INPE).

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
