# Peer review of "Analysis of the Campinas tornado (Brazil) in June 2016: damage track, radar characteristics and lightning observations of the supercell"

_Weather and Climate Dynamics, 2021_

## Referee Comment (RC1)

This study documents some of the first radar observations of a nocturnal tornado event in the austral winter season in the State of São Paulo in Brazil. It gives a meteorological overview of storm, complete with synoptic discussion, overview of observations from several radars and a ground-based lightning detection network, and ground survey of tornadic impacts. While the event is noteworthy and publication is appropriate, the manuscript would benefit from efforts to sharpen the analysis, clarify the meaning of results in the context of the broader literature, and refine the logical support used in several arguments to motivate the study and interpret results. I recommend that the authors address these areas with additional analysis and major revision to the manuscript prior to publication. Related comments and specific suggestions are provided below.

Comments on broader issues:

1. Throughout, the authors have omitted a number of important foundational studies directly related to the topics of interest here. Specific examples include but are not limited to extensive background on the definition supercell thunderstorms (i.e., they are defined and set apart from other intense isolated convection as possessing a characteristic mesocyclone, e.g., Lemon and Doswell, 1979), deeper research into the operational implementation of the Enhanced Fujita scale in the United States with which several key comparisons are made, and considerable background in the area of storm electrification through the non-inductive charging mechanism (e.g., Takahashi 1978, Reynolds et al. 1957, Saunders et al. 2006, among others), physical relationships between lightning and convective intensity, and the lightning jump (e.g., Gatlin and Goodman 2010; Schultz et al. 2009, 2015, 2017). Additional notes are included below in specific comments but these should be considered as broad, substantial topics requiring further support from the literature.

2. A separate but related issue is adequate referencing of instruments and techniques at the first mention. For instance, BrasilDAT isn't thoroughly discussed until section 3.5 but it is mentioned earlier in the paper. Please add references for the instrument at the time it is introduced. Similarly, TITAN is introduced early on without documentation. Additionally, the concept of a lightning jump is described quite a bit in advance of the first citations of the foundational studies behind the concept.

3. I can appreciate the difficulty in thoroughly addressing a storm with limited radar resources, particularly when certain radar fields and variables are either unavailable or unusable because of errors. However, a good deal of the argument and discussion herein refers to "supercell" thunderstorms without proving, or at least convincingly supporting, the presence of a mesocyclone with quantitative rotation speed characteristics (e.g., Lemon and Doswell 1979, Brandes 1984, Moller et al. 1994). For instance, how did the authors infer that the storm discussed in line 264 became a supercell without Doppler velocity data (line 275?). Radar reflectivity characteristics can support the possibility that the storm was a supercell but I caution the authors against overextending the available data. This can still be considered a valuable case study by evaluating the radar reflectivity and lightning parameters but there is no benefit gained from approximating supercell characteristics or onset of supercell development without thorough analysis of Doppler velocity data.

4. I find that aspects of the lightning analysis need to be addressed more carefully prior to publication, discussed further below.
   a. For instance, CG flashes are made up of a number of return strokes that are often reported separately in "stroke-level" CG data. If the term "stroke" is used synonymously with

"flash" herein, please be careful to clearly specify that you are not referring to the numerous return strokes that make up each individual flash.

b. The lightning jump began as a concept in the 1980s that has since been fit with an operational, quantifiable definition. Recent work by Schultz et al. between 2009 and 2017 have clarified a quantitative definition based on the change of flash rate over time (delta flash rate/delta time) that can be easily applied to total lightning data. The application of the concept of "lightning jump" seems to have been applied quite loosely here and would benefit from more thorough quantitative treatment without too much effort given the availability of the total lightning data. Moreover, it appears as though in Fig. 12 that best documents lightning flashes associated with the case, there were actually several jumps as defined through the change in flash rate over time, near 0105 UT, 0220 UT, as well as 0310 UT. It's quite possible that the greatest jump occurred near 0314 UT as discussed in the paper, but this really should be quantified and discussed in the proper context.

c. Related, substantial work has been done to address the physical meaning of lightning jumps in the context of severe weather (e.g., discussion in lines 470-474), including tornadoes (e.g., Schultz et al. 2015, 2017) and even with respect to supercell mesocyclone development and intensification (e.g., Stough et al. 2017). The results from this study are quite consistent with and could be connected better with those in the literature, ultimately bolstering the main conclusions of this paper as they are drawn throughout Section 3.5.

Specific comments:

1. Lines 15, (lines 103-104 as well): Could you please elaborate on your reasoning for limited fatalities during this nocturnal event? Typically, nocturnal tornadoes have been linked with greater human casualties (e.g., Ashley 2007, Kis and Straka 2010).

2. Line 57: The statement that no link has been found between climate change and tornado frequency is misleading at best. Please either find references that support this assertion or refer to studies such as Brooks 2013 to revise and remedy.

3. Lines 58-62: This is admittedly a complex topic to address briefly, but should at minimum, include a few additional ideas such as (a) impacts of microphysical modification from aerosols on buoyancy and (b) competition for water vapor that inhibits drop growth and deep convection.

4. Line 195: Are there any reports or news articles that can be referenced for these figures?

5. Figure 4: This is a nice overview. Could you please somehow annotate the storms of interest to help the reader with context?

6. Lines 277-279: Did the storms truly dissipate (verified through another radar to the NE of SRO) or is it possible that the distance from SRO was causing the storms to be sampled at high altitudes, apparently diminishing their reflectivity characteristics?

7. It would be a great help to include the reference radar in each of the figure captions where radar markers aren't visible in any panel (e.g., Fig. 7).

8. Lines 341-347: Especially for non-expert readers, please explain why these radar limitations may have resulted in underestimates of the quantities of interest.

9. Line 363: Please explain how probability of hail and Foote Krauss category quantities calculated, even just briefly if the authors elect to refer back to TITAN documentation.

10. Section 3.4: The overview and figures of damage here are a nice addition to the paper. However, there are some fundamental flaws in the reasoning and logic used in evaluating the damage in the context of the EF scale and comparisons with United States structures. I think the authors'

EF scale assessment is in the right neighborhood but I disagree with claims that comparisons are difficult between countries because of construction or lack of parallels in the EF scale. Please see comments for more detail.

    a. First and foremost, the EF scale has a number of damage indicators (DIs) used regularly in operations. It's true that it's difficult to rate the strongest tornadoes in areas where vegetation is limited or homes are of poor construction quality. Simply put, there aren't enough indicators that meet the criteria of withstanding the strongest winds. However, this is not a limitation of the EF scale and it should apply quite readily to other regions.  I would encourage the authors to search the literature for applications of the damage scale in rating tornadoes (e.g., Burgess et al. 2014) and/or to reach out to the US National Weather Service to learn more.

    b. Second, please find a reputable reference to support claims on structural composition and integrity such as those in line 371. Census data may be a good place to start, there may also be some other published studies on satellite remote sensing to get a better sense of the geographic distribution of structures. Even in the US, the type and quality of home construction can vary quite a bit by region and it would be best to avoid overgeneralizing. Brick and concrete structures can be extensive in parts of the US, but anecdotally, concrete block structures don't withstand convective winds as well as a solid frame home with brick veneer. I'm also unsure of an earlier claim that motor homes (typically thought of as recreational vehicles) are common and would need a reference to back this up as well (line 47 in the abstract).

11. Lines 404-405: How is optimal coverage defined here? Spatial extent, detection efficiency? Please elaborate.

12. Line 406: Have the authors completed a sensitivity analysis of the reflectivity threshold used to properly associated lighting flash counts with the cell of interest? It can be difficult to identify a repeatable, objective method for flash counting. However, anecdotally, restricting analysis to 30 dBZ often results in undercounting flashes and can result in an artificially low flash rate.

13. Line 440: Please note that IC and total lightning flash rates in excess of 110 min$^{-1}$ are not uncommon in deep convection, particularly in supercells. Suggest removing this reference or adding others that discuss supercell flash rates more extensively.

14. Figure 12 and related discussion: How much of the variability in flash rate and the qualitative jumps can be attributed to cell mergers? Similarly, can any of the decreases in flash rate be associated with cell splits? Please discuss any efforts to control for this effect on the data briefly.

15. Line 469: Please reference a few specific related case studies.

16. Line 550: Could the authors please explain a little more about these difficulties? What tools are recommended generally to remedy the issues discussed throughout this section?

References:

Ashley, Walker S. (2007). Spatial and temporal analysis of tornado fatalities in the United States: 1880–2005. *Weather and Forecasting 22*(6). 1214-1228.

Brandes, E. A. (1984). Vertical vorticity generation and mesocyclone sustenance in tornadic thunderstorms: The observational evidence. *Monthly Weather Review, 112*, 2253–2269.

Brooks, Harold E. Severe thunderstorms and climate change (2013). *Atmospheric Research 123*. 129-138.

Burgess, Donald, et al. (2014). 20 May 2013 Moore, Oklahoma, tornado: Damage survey and analysis. *Weather and Forecasting 29*(5). 1229-1237.

Gatlin, P. N., & Goodman, S. J. (2010). A total lightning trending algorithm to identify severe thunderstorms. *Journal of Atmospheric and Oceanic Technology*, *27*(1), 3–22. https://doi.org/10.1175/2009JTECHA1286.1

Kis, Amanda K., and Jerry M. Straka (2010). "Nocturnal tornado climatology." *Weather and Forecasting 25*(2), 545-561.

Lemon, L. R., & Doswell, C. A. (1979). Severe thunderstorm evolution and mesocyclone structure as related to tornadogenesis. *Monthly Weather Review*, *107*(9), 1184–1197. https://doi.org/10.1175/1520-0493(1979)107<1184:STEAMS>2.0.CO;2

Moller, A. R., Doswell, C. a., Foster, M. P., & Woodall, G. R. (1994). The operational recognition of supercell thunderstorm environments and storm structures. *Weather and Forecasting*, *9*(3), 327–347. https://doi.org/10.1175/1520-0434(1994)009<0327:TOROST>2.0.CO;2

Reynolds, S. E., Brook, M., & Gourley, M. F. (1957). Thunderstorm charge separation. *Journal of Meteorology*, *14*(5), 426–436. https://doi.org/10.1175/1520-0469(1957)014<0426:TCS>2.0.CO;2

Saunders, C. P. R., Bax-norman, H., Emersic, C., Avila, E. E., & Castellano, N. E. (2006). Laboratory studies of the effect of cloud conditions on graupel/crystal charge transfer in thunderstorm electrification. *Quarterly Journal of the Royal Meteorological Society*, *132*(621), 2653–2673. https://doi.org/10.1256/qj.05.218

Schultz, C. J., Petersen, W. A., & Carey, L. D. (2009). Preliminary development and evaluation of lightning jump algorithms for the real-time detection of severe weather. *Journal of Applied Meteorology and Climatology*, *48*(12), 2543–2563. https://doi.org/10.1175/2009JAMC2237.1

Schultz, C. J., Carey, L. D., Schultz, E. V., & Blakeslee, R. J. (2015). Insight into the kinematic and microphysical processes that control lightning jumps. *Weather and Forecasting*, *30*, 1591–1621. https://doi.org/10.1175/WAF-D-14-00147.1

Schultz, C. J., Carey, L. D., Schultz, E. V., & Blakeslee, R. J. (2017). Kinematic and microphysical significance of lightning jumps versus nonjump increases in total flash rate. *Weather and Forecasting*, *32*(1), 275–288. https://doi.org/10.1175/WAF-D-15-0175.1

Stough, S. M., Carey, L. D., Schultz, C. J., & Bitzer, P. M. (2017). Investigating the relationship between lightning and mesocyclonic rotation in supercell thunderstorms. *Weather and Forecasting*, *32*(6), 2237–2259. https://doi.org/10.1175/WAF-D-17-0025.1

Takahashi, T. (1978). Riming Electrification as a Charge Generation Mechanism in Thunderstorms. *Journal of the Atmospheric Sciences*, *35*(8), 1536–1548. https://doi.org/10.1175/1520-0469(1978)035<1536:REAACG>2.0.CO;2

---

## Referee Comment (RC2)

Review: Analysis of the Campinas Tornado (Brazil) in June 2016: damage track, radar characteristics and lightning observations of the supercell

Author: Lucí Hildago Nunes, Gerhard Held, Ana Maria Gomes, Kieber Pinheiro Naccarato, and Paul Reis Amorim

Paper ID: wcd-2021-35

**Synopsis:** The authors analyze a tornadic supercell that struck Campinas in southern Brazil on the night of 05 June 2016. They discuss the synoptic situation and then dive into the analysis of the storm itself using three radars, emphasizing one, a total lightning network and the damage left behind. As they discussed the main supercell, the authors briefly mentioned another tornadic supercell that struck nearby but no significant analysis was done.

**Overall Comments:**
- I find the writing in the introduction to be haphazard, shifting from one subject to another, even in the same paragraphs. The topics need to be streamlined. For example, I found that section 3.5, damage, should be the final section in the paper and should come after 3.4, lightning. There are other sections where a topic shifts to another topic and then returns to the same topic in a confusing manner. I've described these in the specific comments.
- I left the introduction not knowing what this paper will cover or its objectives and goals. The introduction should specify what this manuscript will cover, how each part relates, or builds upon previous parts. Also, the introduction should set my expectations on the scope.
- Where's the sounding thermodynamic and hodograph analysis? The authors describe the overall synoptic setting and then go straight to radar, lightning and damage analysis, without discussing the near storm environment. I would've expected a discussion of a near storm environmental analysis including vertical thermodynamic and wind profile analysis given the critical role it plays in assessing the threat posed by storms, in addition to basic storm dynamics, behavior and motion. It seems the authors are perplexed by the storm motion, and I'm not surprised with no discussion on the vertical profile of the wind and thermodynamics. At least discuss the near storm environmental values including vertical shear, mixed layer CAPE and CIN, storm-relative helicity, lifted condensation levels for a couple points along the storm's path. A supercell lasting 8.5 hours most certainly encountered a gradient of environmental parameters. That the main tornado occurred after 0300 UTC suggests an interesting shift between increasing CIN and increasing low-level shear may have occurred (see Bunker et al. 2019).
- The above concern leads to another concern about the author's attribution of relatively low reflectivities in the storm core to the season. Well that may be true but that alone is an insufficient reason and I think comparing the above near storm environment to the more traditional convective season cases may allow for a little more understanding than the none that I see in the text.

**Specific comments:**

Abstract:

- Line 16-17: "The affected areas are middle and even upper-middle class neighborhoods, with solid buildings, confirming the potency of the phenomenon. " I'm not sure a phenomenon can be classified as potent by where it hits. The following sentence provides a more compelling reason to call this event potent.
- Line 34: "…with a "lightning jump" from 0 to 55 ground strokes per minute …" I mention this in the lightning section but the lightning jump, as described by Schultz et al. (2009) represent total lightning.
- Line 37-39: "…this is most likely due to the fact that this was the first occurrence of a tornado observed by radar during the dry austral winter season in this region of Brazil, as well as a nocturnal event." I describe my concerns about making a connection without further discussion in the rest of the paper.

Introduction:

- Line 41: The authors state: "Tornadoes are the most intense vortices in the atmosphere over land, formed in environments with intense wind shear. " They can form in environments without intense wind shear. However most do. Thus state that they form mostly in environments exhibiting strong vertical wind shear.
- Line 42-43: The authors state: "They are associated with conditions of great thermodynamic instability, reinforced by specific surface parameters, such as relief, vegetation, urbanization and the presence of water bodies. " Thermodynamic instability, as measured by mixed layer CAPE, only need be sufficient, and can be modest (e.g., $>\sim$ a few hundred j/kg). So it would be best to state sufficient CAPE instead of great thermodynamic instability. Also, I'm not sure what the authors implay when they say they are reinforced by relief, urbanization and water bodies. Sure all of those features modulate the presence of tornadoes but stating such implies that much more will be discussed on the details of this reinforcement.
- Lines 47-48: "in the USA motor homes are commonly used, while in the plains of North America timber buildings prevail " Wood framed houses dominate in most parts of Canada and the USA. Only in FL and a few other areas do concrete masonry unit houses prevail. Meanwhile, motor homes should be manufactured homes. There is an important distinction between the two.
- Lines 50-54: "Tornadic cells can develop in atmospheric environments that produce severe weather conditions such as …" There are a confusing mix of names, some associated with atmospheric boundaries (eg fronts), while others are convective storm modes (supercells). My suggestion is to stick with the variety of storm classifications (e.g., ordinary cells, Quasi Linear Convective Systems, Supercells). Leave the cyclones and fronts out unless the authors want to talk about what generates the parent storms of tornadoes.
- Lines 55-65: The paragraph starting with "The rising number of detected tornadoes…" appears to set the topic of this paragraph to be about the effects of population on tornado

numbers.  But then the subject wonders off to global warming's impacts on tornadoes and then quickly to the effects of aerosols on convective behavior.  Such wondering makes my head spin and confuses me as to where this introduction is going.

- Line 71: "…which are seasonal transition periods with more atmospheric instability." Let's specify what is meant by instability.  Is it that there is more CAPE in the transition seasons or is it more a combination of shear and CAPE?  I would think that it's the latter if we're talking about favorable environments for supercell tornadoes.

- Line 85-86: "Some of the studies also attempted to investigate numerical forecast models …" This is not related to the lead sentence of this paragraph.  Success stories of NWP in forecasting severe weather deserves its own paragraph.

- Lines 89-91: "Since 1994, several events of supercell storms traversing the State of São Paulo while spawning tornadoes, although relatively rare, had been observed and tracked by the Doppler S-band radars…" This paragraph started by introducing several tornadic supercell events crossing São Paulo.  However, the authors crossed through several other topics on climatology of severe weather in São Paulo.  If this paragraph is about the climatology of severe weather in São Paulo, then state that up front.

- Line 105:  The EF Scale replaced the F Scale in the US in 2007 and in Canada in 2013. The F Scale is still used in parts of Europe.

- Line 107:  Please use the link above and cite it as NWI (2006) where NWI is the National Wind Institute.

- Lines 107-113:  This is the paragraph that describes what this study is about.  However I have little direction as to what makes this case study important to study.   Instead, I was exposed to a variety of topics related to severe storms in southern Brazil ranging from radar studies to seasonal climatology.  I will read on to see what the case study is really about.

- Lines 127-128:  "Also, a difference of 10–15 dBZ between BRU and SRO was noted, and therefore the authors decided to only use these data in a qualitative manner " It seems that BRU's reflectivity would be of higher quality.  However, are the authors equally skeptical of BRU's calibration?  Otherwise, I understand the author's concern for comparing specific reflectivity values between both radars.

- Line 141:  "… speed and direction of propagation, etc, per volume scan, as well as cell tracking, including splits and mergers of cells. " Perhaps instead of 'etc,' mention that TITAN produces volume scan products relevant to this study that include Area, Volume, Precipitation Flux, VIL, Maximum Reflectivity, hail metrics, storm motion, as well as cell tracking capable of monitoring splits and mergers?  I'm assuming speed and direction of propagation is at the storm cell level but correct me if I'm wrong.

- Line 153:  "…such as regions where opposing radial velocities along the azimuth are observed,…" I'm not sure what along an azimuth implies other than radial divergence/convergence.  Instead, the text should read as localized regions of strong azimuthal shear, or radial velocity couplets whose velocity minima (maxima) are located roughly azimuthally adjacent to each other.  I prefer the former.

- Lines 154-155: "A tornado will be spawned when the cloud funnel extends to the ground, where it will create a characteristic damage pattern, which in turn will reveal the magnitude of the phenomenon. " A tornado need not visualize itself as a funnel cloud

extending to the ground.  In this paragraph, stick to the radar-based indications of tornado signatures and tornado vortex signatures.  See papers by Rodger Brown and Vincent Wood.  Start with this (https://doi.org/10.1175/WAF-D-11-00111.1 and go back through their citations.  Then if needed, elaborate on the ground-truth indications of tornadoes in another paragraph.

- o Further elaboration on tornado evolution above.  I suggest reading up on work related to the tornadogenesis evolution in that most tornadoes form at low-levels and then extend upward, or form through a deep layer connected to the surface. Search for articles from Jana Houser
- Lines 182-198: "The tornado of 05 June crossed the city from west to east, …"  This section is describing the event and should have its own topic header.
- Lines 184-185: "...with solid constructions, a fact that attests the severity of the phenomenon."  What is the severity of the phenomenon.  The introduction only mentions that this was an F(EF)3 tornado but does not mention where this damage occurred, and certainly not relative to the middle class neighborhoods.  We
- Lines 196-199: "...74 mm of rain were recorded in just 45 minutes …"  I think after the authors expand upon the impacts, explain on the goal of this paper in the intro, there may not be time to discuss the rainfall aspects of this case.

Synoptic Situation

- Lines 204-206: "The consequential circulation advected moist air from the Amazon and Pacific region throughout the troposphere, resulting in a baroclinic flow with unstable conditions (Fig. 2) favorable for the development of severe thunderstorms in the State of São Paulo, even during the night."  I'm not sure how the 250mb anticyclone is advecting any moist air relevant to the thunderstorms.  It is the flow in the lower third of the troposphere that is advecting the majority of the moisture utilized by convection.  Going with the archive (http://tempo.cptec.inpe.br/boletimtecnico/pt), the 850mb level shows a tight channel originating from NW Brazil and that is easily a plausible source for the moisture advection.  Further, this passage seems to imply that the moisture advection is resulting in a baroclinic flow.  Did the authors really mean to say what they did?  Perhaps they mean to state that the west side of a deep anticyclone over central Brazil resulted in a low-level jet that transported moisture to the ESE and into a zonally oriented baroclinic zone across the target area.  The CAPE generated by the moisture encountered lifting along the front and resulted in deep convection.  Figure 3 should include the 850 mb chart.

Radar Observations

Overview of Severe Convective Activity on 04/05 June 2016

[Figure]

- Line 228: "The severe convective cells (≥35 dBZ) …" This is a low bar for considering cells severe. How about just 'convective cells'.
- Lines 242-244: "Campinas region can certainly be classified as a supercell, based on several of its characteristics, such as velocity, echo tops penetrating the tropopause, as well as "severe storm parameters" generated by the TITAN analysis " The requirements for a convective cell to be classified as a supercell are really storm updraft rotation and persistence (see the definition of a supercell in the AMS: https://glossary.ametsoc.org/wiki/Supercell) So, the description here could be modified to say that this storm can be classified as a supercell based on its rotation and longevity. It also had a storm top above the tropopause, and exhibited several storm attributes associated with with severe weather.
- Line 266: "At around 00:30 UT, a new cell had suddenly developed …" Adding labels in figure 6 and referring to them here may help the reader.
- Line 268: "passing over São José dos Campos) " I cannot find it in Fig. 6.
- Line 291: "All these major storms were typical multi-cellular complexes, …" I'm not sure I understand after the authors were describing these as supercells. Please elaborate.
- Line 299: "…had reached the ground,…" The authors don't know if the vortex descended or ascended. They only know the tornado began about this time.

Details of the Campinas Supercell and Tornado

- Lines 300-306: "These parameters indicate 305 extremely strong convective activity, considering the fact that this storm occurred during the generally dry month of June. "There are several features of this storm that could've been analyzed but not mentioned. Was there a strong echo overhang, or a BWER? Also, there are multiple citations to previous publications, of which a few could be cited. The first place to attend to is Moller's chapter in the AMS monograph 'Severe Convective Storms' (https://bookstore.ametsoc.org/catalog/book/severe-convective-storms).
- Line 311: "($|V_{in}| + |V_{out}|$)/2 " While this may work with storm-relative radial velocities, mesocyclones may appear with a max and min velocity of the same sign. The above

relation may underestimate rotational velocity in those situations. a more generalizable version is (Vmax – Vmin)/2

- Line 318-319: "The rotational vorticity can be calculated from the ratio between the speed and the distance between the pair of opposing radial velocities, yielding a value of $2.5 \cdot 10^{-3}$ s$^{-1}$ for this case. " This is fine but two things: one is that rotational vorticity is not something that one radar can calculate and so the term 'azimuthal shear' is more accurate, second, nothing more is done with this value after this sentence. Thus I suggest removing this sentence unless it is put in perspective.

- Lines 319-321: "From studies of tornadic storms in the USA, threshold values of $V_r \geq$ 12.5 m s$^{-1}$ within a radius of 150km, and $\geq 8.5$ m s$^{-1}$ for distances further than 150km were defined (NSSL, 1985)." There are much more comprehensive studies of rotational velocity and relationships to tornado probabilities in the decades since the NSSL publication. The authors should refer to them rather than an article that had such a small sample size. More recently, papers by Smith et al. (2020a,b), Gibbs and Bowers (2019) have detailed tornado probability statistics vs rotational velocity and environmental parameters (e.g., significant tornado parameter) from thousands of events. The authors should note that the radars they used have different beam widths than the WSR-88D network. NSSL (1985) depended on 1 deg beam width with roughly 5 minute intervals and the more recent citations used super resolution 1 deg beam width with half degree overlap. Thus this example will make for some challenge to compare to US cases directly except to say that wider beam width is likely to lead to lower rotation velocities as papers by Brown and Wood discussed. I don't see any mention of the beam width from the BRU radar but any comparisons to other studies should include a comparison of beam width and other relevant radar characteristics.

- Lines 352-354: "…but also it occurred during the meteorological transition phase of austral autumn, while the current case happened during the dry winter season, when convective cells are less intense. " Stating that because a storm occurred during a dry season is not sufficient to explain why it's attributes are less impressive than another storm that occurred in the wet season. In ingredients based analysis, look at the more direct causes behind the behaviors with the storms that resulted in the differences in the impacts.

- Lines 354-357: "Figure 8 also shows that the speed with which the tornadic cell propagated throughout its lifetime varied between 50-65 km h$^{-1}$, which is characteristic for rapid new cell development in the immediate vicinity, and especially ahead of the mother cell. " I'm not debating how new cell development is affecting its motion. But since this was classified a supercell, why wasn't there an analysis of the mean convective layer wind, and supercell left and right motion vectors. I suggest seeing Bunkers (2018) and his earlier works. The most recent article dives into some of the errors into the supercell motion techniques widely in use by many agencies today. Perhaps some of those error sources are related to the multicellular behavior of many supercells. At least discuss the possible sources of anomalous motion but please start with assessing the supercell motion vector in the near storm environment and compare.

- Line 355: "…identified in Fig. 5 with x…" The figure 5 caption refers to x as a place name. Is this what they're referring to here?

- Line 360: "…but FOKR (Foote Krauss) category 2 (of 4) only. " What is FOKR? Cite a reference.
- Line 364: "…the "HailMassAloft" reached a maximum of 10.8 ktons." Again, is there a reference. At least when introducing TITAN, mention this as another attribute.

Damage

- Line 371: "…American frame…" American wood frame. There are steel framed homes.
- Line 373-374: "Therefore, the degree of destruction in Campinas highlights both the destructive power of the phenomenon and the difficulty of adapting the EF scale to the Brazilian case." If this is so, then how was the rating determined, as stated in line 370?
  - Related to the above question is a comment. There are elements of the brick and cement-walled houses that share similarities to US-built wood-framed homes. The most obvious one is that the roofs appeared to be constructed from wood trusses, rafters and perhaps decking. Figure 9e and f appeared to show that the roofs suffered anomalous damage compared to the rest of the houses and that may mean the roof-to-wall connections were similar to US homes and that this is where the EF Scale can be used with more confidence.
- Lines 380-383: This paragraph is not related to damage and should be in a different section. Or the section could be renamed to 'verification'. Perhaps it's best to discuss determining what type of hazard hit in one section and then more details on the damage in another section.
- Lines 387-389: "Although this area is outside the Bauru radar coverage, the evidence strongly suggests that it also was a tornadic cell, as can be seen in Fig. 10, with metal structures twisted around light poles, concrete poles twisted, large trees uprooted, as well as widespread damage to homes and commercial buildings. " The evidence discriminating tornadic from nontornadic thunderstorm wind damage is a damage swath that is relatively long and narrow with substantially more intense damage than adjacent areas. However, damage > EF2 is more likely to be associated with something other than thunderstorm induced straight-line winds as most straight-line wind events happen under EF2 strength (Edwards, 2018). However, the damage pattern should still be nondivergent and preferably extending a long axis (aspect ratio > 4). Regarding large trees, being a marker for tornadoes, they are not. The probability of tree-fall for a give wind speed actually goes up as tree trunk diameter increases (e.g., Peterson, 2007). Damage greater than EF1 has been associated with significant loss in NDVI as reported by Molthan et al. (2014).
- Line 393-394: "Based on the massive destruction, the phenomenon could be rated as an EF3 tornado. " I suspect that there is no official rating and that the authors are suggesting one. In that case, there should be more specificity as to why the EF3 rating should be applied. Use NWI (2006) and adapt to the changes in wind resistance. But at least point toward the reference and point to the documentation, either in Fig. 10, a citation, or a revised version of the figure. I note that Almeida and Lombardi, 2019 have satellite imagery showing extensive denuding of trees.

- Section 3.5 Lightning Observations:  All of the text concerning the lightning network and its attributes should go into the data and methods section.  I say the same for how the TITAN storm tracking software works.  Leave the results here.

Lightning

- Line 435:  "The third peak evidences a typical *lightning jump"* Please cite the term lightning jump when first used. I see there are citations later on in this section.  Better yet, don't mention lightning jump until you're ready to around line 445.
- Lines 445-446:  "The large number of strokes produced by the event (Fig. 12) has to be highlighted, particularly the *lightning jump* effect, which is a very rapid intensification of the electrical activity inside the thundercloud …"  This is awkward wording.   Why not word this as " The large increase in the lightning discharge rate produced by the event (Fig. 12) represents a lightning jump, which is a very rapid intensification of electrical activity inside the convective cell…"
- Line 449:  "…and downbursts or microbursts." Just mention downbursts.
- Line 451:  "The updrafts together with the gravitational forces…" The physical processes could be shortened
- Line 458-474:  "Figures 13 and 14 depict the tornadic cell…" This paragraph appears to be a discussion on the CG lightning activity that needs some help.  The most important would be to have a first sentence that tells the reader what this upcoming discussion concerning the CG lighting is going to be about.  If I were to headline this discussion it would be that the authors observed trends in CG lightning frequency that differed from earlier studies.  The content of the paragraph could use some help in readability.  I would then describe the pretornadic phase, then the tornadic phase.  The authors jump from pretornadic to tornadic then back to pretornadic then tornadic.  Perhaps reorganize the figures so that figure 13 is pretornadic and 14 is tornadic phases and showing lightning, reflectivity CAPPI and reflectivity vertical cross section.
- Lines 470-471:  "during the assumed touch-down of the tornado…" The authors don't know if the vortex intensified down near the ground first, or at higher elevations first.  Remove the word 'touch-down' and just say tornado formed.
- Line 474:  "…but it might be attributed to the fact that this tornadic storm was the first of its kind documented by radar during the dry winter season. "  I am uncomfortable with implying there is any relationship between the unusual  CG frequency changes (pre-tornadic to tornadic) with what season the event happened to occur.  This is a single case study and not a statistical study of thunderstorm behaviors.  The US cases of CG lightning trends have shown high variability with poor case to case consistency and that's why the total lightning trend was more salient.
- Lines 479-485:  This paragraph reads more like a figure caption.  In fact, a lot of it is repeated in the caption.  I suggest reorganizing the figures as mentioned above and removing any content that reads like a figure caption.

Summary

- I was hoping the summary would clear up the difference between thte maximum reflectivity observed from 55 to 65 dBZ in line 511 and then the 49.5 dBZ maximum reflectivity observed in line 515.  But I'm still unclear where the disparity is originating from.
- Line 516-517:  "…certainly an underestimate due to the long radial distance, but possibly also because this was the first tornadic event ever recorded by radar during the austral dry winter period …" Back to the seasonality of convection, I think just stating that because the storm occurred in the dry season the peak reflectivity and VIL are lower is not a sufficient explanation.  What environmental parameters were different with this event than with the other events with higher reflectivity?  In the US, there are numerous studies on the differences between cool season convection and warm season convection, and diurnal to nocturnal events.  I suggest the authors dive into the works by Bunker et al. (2019) and Hanstrum et al. (2002).  Finally, how does it really matter whether or not the reflectivity is less or the VIL is less?  The answer derives from what hazards are being predicted.  If hail size is of concern, then perhaps it does to some extent.  However the inferences I get from this paper is that the tornado threat is the primary concern.  If so, then the maximum reflectivity and VIL is of little relevance and will provide little skill in short term tornado prediction and warnings.  It will be the trends in updraft strength, mesocyclone strength, and the near storm environment related to tornadoes, that provide the best information.
- Lines 520-523:  "Other typical tornado signatures, such as a rotational damage pattern of uprooted and broken trees, as well as a hook echo and a mesocyclone with a rotational -1 velocity of 12.5 m s , both observed in the radar data at the time of the assumed spawning of the tornado (Volume Scan 03:14-03:22 UT)"  This sentence contains signatures that only can be evaluated after the fact with those that occur before or during the event.  Remove any content involving signatures that can only be evaluated after the event.
- Lines 533-534:  "…touch-down time of the tornado…" Remove 'touch-down' and replace with 'formation'.
- Lines 543-545:  "According to eyewitnesses, another tornado spawned in the same region from a different supercell some three hours before the one that impacted Campinas, traveling on a parallel track. Based on the damage pattern in the small town of  Jarinu, it could be considered as an EF3 tornado."  Up to this point, the summary was discussing the main supercell that passed over Campinas, or a comparison of this storm with past events.  This sentence does neither.  Either remove or discuss why this is relevant.
- Lines 546-550:  "Considering the relatively rare occurrence, small-scale features and short duration of tornadic cells in the central region of the State of São Paulo, it is almost impossible to predict a possible formation of tornadoes timely for issuing warnings to the population. …" I beg to differ.  First of all, tornadoes are rare phenomena anywhere, however severe weather environments are relatively common in southern Brazil compared to many other areas frequently exposed to convection (e.g., Brooks et al. 2003).  Given a favorable enough environment for tornadoes , just identifying a certain cell as a supercell can be sufficient for expecting a tornado and being able issue a warning (see Smith et al. 2015, Krocak et al. 2021).  Perhaps early supercell identification may be more difficult, but not impossible.  Even if detection comes at a

later stage, a tornado forecast is possible for areas ahead of the storm.   I think the real challenge is getting enough tornado verification to provide a reliable performance skill to be able to test the performance of issuing valuable short term tornado forecasts and warnings.  Therefore, I strongly suggest that instead of saying it's almost impossible to predict, that further work is needed to create verification statistics from improved tornado reporting.

- Lines 553-554: "Furthermore, the currently available Doppler radars in Brazil are insufficient for a countrywide coverage to detect severe storms in real time for issuing timely warnings. "  I'm not convinced that the evidence provided here supports such a conclusion generally across south Brazil.  I do agree the radar coverage prevents detailed storm interrogation for accurate warnings in many areas.  But this paper does not provide anything but one anecdote.  The authors will need more to make a defensible claim such as this.  As stated above, given an appropriately favorable tornado environment, and an identification of a supercell, a tornado warning can be justified.  But there was not a sufficient analysis of the environment.  But there was sufficient radar to identify a supercell.  Again, I agree a denser radar network would certainly provide more information about low-level characteristics of storms to provide better short term tornado forecasts.  But again, there needs to be a more general statistical analysis involving more cases.

But I'm not sure whether the near storm environment in this case study is appropriately favorable for tornadoes because there was no analysis of the vertical thermodynamic and kinematic structure of the atmosphere.  A near storm Skewt and hodograph analysis was not even presented in this paper.

Technical comments:

- Line 33: "12.5 m s-1 " should be 12.5 m s$^{-1}$
- Line 51: "50 m s-1 "  should be 50 m s$^{-1}$
- Line 125: "VOL-scans…"  volume scans
- Line 126: "…individual rays…"  individual radials
- Line 176: ", this program..." swap comma for period  " . This program…"
- Line 179: "…or shelters to affected people"   "…or shelters for affected people"
- Line 180: "…false sense of resilience to the public …"  "…false sense of resilience among the public…"
- Line 238: "00:52 LT"  add UT time too.
- Line 271: "…moved at ±60 km h$^{-1}$  …"  remove the ±, also, "towards east-northeast "  rephrase to "toward the east-northeast"
- Line 272-273: "…during 40 min until reaching a peak at 00:50 UT …"  rephrase to "during the 40 minutes prior to reaching its peak at 00:50 UT."
- Line 320: "…within a radius of 150km…"  change to "…within a range of 150 km…"

- Lines 332-333: "Echo Top (10 dBZ contour); Max Reflectivity (dBZ); VIL (kg m$^{-2}$) and propagation velocity of the 35 dBZ echo cores. " Commas should be sufficient to separate these items.
- Line 337: "radial distance" range.
- Lines 342-344: "Their observed values of 49.5 dBZ and 15.8 kg m , respectively, are certainly an underestimate, but considerably lower than those recorded…" It seems that the word 'and' should replace 'but'.
- Fig. 8 right Y axis label: VIL, speed, height are not units.
- Line 370: "A field recognition conducted along the storm path …" A field reconnaissance. Preferably, a damage survey is another term to use.
- Line 403-404: "…EarthNetworks technology…" A trademark may need to be applied to EarthNetworks
- Line 502: "…Doppler S-band adars…" radars

References
- The TORRO URL cannot be found
- Line 686: The reference Insse does not appear in the text
- Line 714: The better reference is NWI (2006). NWI=National Wind Institute

References cited in this review:
- Bunker et al. (2019): https://doi.org/10.1175/WAF-D-18-0162.1
- Brooks et al. (2003): https://doi.org/10.1016/S0169-8095(03)00045-0
- Bunkers (2018): https://doi.org/10.1175/WAF-D-17-0133.1a
- Edwards et. al. (2018): https://doi.org/10.1175/JAMC-D-17-0306.1
- Gibbs and Bowers (2019): https://doi.org/10.15191/nwajom.2019.0709
- Hanstrum et al. (2002): https://doi.org/10.1175/1520-0434(2002)017<0705:TCSTOC>2.0.CO;2
- Kingfield and LaDue (2015): https://doi.org/10.1175/WAF-D-14-00096.1
- Krocak et al. (2021): https://doi.org/10.1175/BAMS-D-19-0310.1
- Molthan et al. (2014): http://dx.doi.org/10.15191/nwajom. 2014.0216.
- NWI (2006): https://www.depts.ttu.edu/nwi/Pubs/FScale/EFScale.pdf
- Peterson, C.J. (2007): https://doi.org/10.1016/j.foreco.2007.03.013
- Schultz et al. (2009): https://doi.org/10.1175/2009JAMC2237.1
- Smith et al. (2015): https://doi.org/10.1175/WAF-D-14-00122.1
- Smith et al. (2020a): https://doi.org/10.1175/WAF-D-20-0011.1
- Smith et al. (2020b): https://doi.org/10.1175/WAF-D-20-0010.1